# Job demands, job resources and postdoctoral job satisfaction: An empirical study based on the data from 2020 *Nature* global postdoctoral survey

**Yue Zhang, Xinxing Duan** *

School of Public Policy and Management, China University of Mining and Technology, Xuzhou, China

* xxduan@163.com

**Data Availability Statement:** The raw data of Nature's Global Postdoc Survey are available from figshare at go.nature.com/3tmckuq. The relevant

## Abstract

Postdocs encounter numerous hurdles in terms of their professional survival and academic development, as a result of institutional reform and the prevailing academic environment. These challenges significantly impact their job satisfaction, which in turn plays a crucial role in shaping their scientific research career trajectory. To facilitate the advancement of relevant systems and augment the job satisfaction of postdocs, this study employs the 2020 *Nature* Global Postdoctoral Survey data to conduct a comprehensive analysis. Utilizing descriptive statistics, correlation analysis, and regression analysis, the impact of job characteristic elements on postdoctoral job satisfaction was examined within the theoretical framework of the Job Requirements-Resources (JD-R) model, as well as the mechanisms by which job characteristic elements impact postdoctoral job satisfaction. It was found that job demands and job resources negatively and positively predicted postdoctoral job satisfaction, respectively, with job burnout and job engagement playing a partial mediating role. Job demands can drive postdocs to develop negative coping psychology and limit the motivating effect of job resources on job engagement; job resources can act as a buffer to reduce the probability of postdocs experiencing job burnout as a result of job demands. The aforementioned findings generally support the applicability of the JD-R model to postdocs, theoretically revealing the intrinsic psychological mechanisms by which job characteristics influence postdoctoral job satisfaction and providing theoretical supplements and practical references for postdoctoral training and management.

## Introduction

In the second half of the 19th century, the scientific and industrial revolutions drove tremendous growth in productivity, and science and technology became the determinants of economic growth. Under the internal dynamics of science's complexity and the external influence of its utilitarian value, scientific research and knowledge production became more institutionalized and organized, and the demand for high-level researchers grew rapidly [1]. In this context, the postdoctoral system, which concentrates on enhancing the research capacity of young

data used in the study are within the paper and its Supporting information files.

**Funding:** This study was supported by the Fundamental Research Funds for the Central Universities(2023XSCX033), the Major project of Philosophy and Social Science Research in Colleges and Universities of Jiangsu Province (2020SJZDA121), the Postgraduate Research & Practice Innovation Program of Jiangsu Province (KYCX23_2622), the Graduate Innovation Program of China University of Mining and Technology (2023WLKXJ127), and the Research and Practice Project on Graduate Education and Teaching Reform of China University of Mining and Technology(2023YJSJG016). The funders had no role in study design, data collectionand analysis, decision to publish, or preparation of the manuscript.

**Competing interests:** The authors have declared that no competing interests exist.

physicians, emerged and expanded globally. Today, the postdoctoral system has become an important system for cultivating high-level innovative young talents, and the development of the postdoctoral group, as a professional engaged in scientific research, has become an important academic reserve army for the mission of scientific research and innovation, attracting the attention of governments and scholars [2]. In recent years, the number of postdocs has increased exponentially in all countries, with the scope of postdoctoral entry expanding in all nations. However, the rapid growth of postdocs is frequently accompanied by a disregard for quality assurance and individual job satisfaction [3]. In 2020, Nature conducted a large-scale survey on global postdocs in terms of role status, salary and compensation, working hours, the impact of the COVID-19 epidemic, job satisfaction, racial discrimination, physical and mental health, and job prospects. In terms of job satisfaction, more than a quarter (26%) of postdocs said they were dissatisfied with their current job status; 32% of respondents said their postdoc job was worse than they expected; and more than half of postdocs said their job satisfaction had significantly decreased in the past year. The issue of job satisfaction within the postdocs within the present career environment is intricately linked to the general efficacy of forthcoming research careers. Consequently, the initial stride towards enhancing the condition of the postdoctoral profession involves the identification of the origins of dissatisfaction among postdocs [4]. Current research reveals that postdocs experience high levels of tension and average levels of job satisfaction [5], which are influenced by factors such as personal expectations, organizational support, management systems, and innovation requirements [6]. These findings contribute to our comprehension of the present state of postdoctoral satisfaction and the factors that influence it. However, there remains a need for a more comprehensive explanation and clarification of the relationships between these influencing factors. Additionally, a more robust theoretical framework is required to systematically investigate postdoctoral job satisfaction, its influencing factors, and the underlying mechanisms involved. The Job Demands-Resources (JD-R) Model is a comprehensive framework that considers the impact of both job demands and job resources on mental health in the workplace. This model has gained significant popularity in many occupational field studies due to its versatile and adaptable analytical framework, which takes into account the direct and interactive effects of both factors [7]. Based on this, this study utilizes data from the 2020 *Nature* Global Postdoctoral Survey to investigate the impact of job characteristic elements on postdoctoral job satisfaction and the mechanisms by which job characteristic elements affect postdoctoral job satisfaction under the theoretical framework of the Job Demands-Resources (JD-R) model. Aiming to provide a comprehensive and detailed analysis of the internal mechanisms of postdoctoral job satisfaction and empirical evidence to provide reference for strengthening the postdoctoral system and improving the job satisfaction of postdoctoral researchers.

## Literature review and research hypothesis

### Literature review

**Connotation of job satisfaction.** The concept of "job satisfaction" originates from the field of organizational behavior, and presently lacks a precise and universally accepted definition. There exist two prevailing meanings that are frequently employed in the discourse. The first definition pertains to the state of experiencing pleasure derived from the assessment of a job, wherein such pleasure is contingent upon the recognition of the profession's inherent value. The second definition concerns the degree to which individuals have positive or negative sentiments towards their respective occupations [8]. Furthermore, it is noteworthy to mention that process theory and content theory have significant positions as fundamental theoretical frameworks in the study of job satisfaction. The content theory posits that job

satisfaction is perceived as being relatively stable and is influenced by a range of job-related factors, including but not limited to job achievement, recognition of one's efforts, sense of responsibility, prospects for career progression, support from the organization, working conditions, and involvement in organizational decision-making [9]. Process theory places significant emphasis on the concept of "process", which entails examining the origins of job satisfaction within the dynamic framework of work. The reasons of these outcomes are mostly associated with the exertion of employees, including the degree of effort invested, the regularity of effort exerted, and the adjustments made in response to environmental or personal circumstances [9]. In the realm of postdoctoral studies, postdoctoral job satisfaction pertains to the subjective perceptions held by postdoctoral researchers towards their employment, representing an emotional connection between individuals and their postdoctoral positions. This encompasses both intrinsic happiness derived from meeting employment expectations and anticipating career growth, as well as extrinsic fulfillment derived from favorable academic conditions, interpersonal connections, salary, and social standing [6].

**The current status of postdoctoral job satisfaction.** Numerous nations have researched postdoctoral employment satisfaction. *Science* conducted a survey on postdoctoral work as early as 1999, revealing postdocs' discontent with work pressure and employment situation [10]. However, recent research findings, indicate that the situation has not improved [3]. In 2015, Huazhong Agricultural University conducted a sample survey on China's postdoctoral population and found that 59.2% of postdocs were "generally" or "dissatisfied" with their job status [11]; In 2017, a survey of North American postdocs revealed low levels of satisfaction, with approximately 20% of postdocs losing interest in academic careers and 30% of respondents indicating that they would not recommend others to become postdocs [12]. In 2020, *Nature* published the results of its first global postdoctoral survey, which revealed that only 12% of postdocs were contented with their employment status [13]. In 2021, *Nature*'s Global Compensation and Satisfaction Survey revealed that the average job satisfaction of postdocs was 2.43, which is lower than the median of 3, confirming that postdocs are less likely to be contented with their jobs [14]. Studies also indicate that job satisfaction mediates the relationship between job stress and postdoctoral desertion of academic careers, i.e., increased job stress leads to decreased job satisfaction, which ultimately leads to postdoctoral abandonment of academic careers [15]. In order to prevent the loss of research talent and social interests, it is evident that the issue of postdoctoral job satisfaction requires immediate attention.

**Influencing factors of postdoctoral job satisfaction.** The factors affecting postdoctoral job satisfaction can be categorized into four groups, based on an overview of existing research. The first category consists of material factors, such as salary [16], financial support [2], and welfare benefits [17], which are crucial considerations for the majority of postdocs when they leave the station and enter the job market. The second category consists of security factors, such as training programs [18], career advancement opportunities [19], social support [20], etc. The postdoctoral group has a brief contract period with the university, which is both unstable and developmental, and their survival and production of high-quality results depend on security resources and development opportunities. Life factors, such as overtime work, work hours [21], etc., comprise the third category. Individual factors, such as personal expectations, self-efficacy [22], sense of job accomplishment [23], and mental health [24], comprise the fourth category. In recent years, with the increase in the number of postdocs worldwide, the increase in years of experience, and the increase in pressure, mental health issues have become more prevalent among postdocs, and have also become a significant factor in determining the postdoctoral job satisfaction [15]. From the perspective of the JD-R model, the four categories of factors influencing postdoctoral job satisfaction are consistent with the categorization features of job characteristic elements. Both the first type of material factors and the

second type of security factors can provide support and assistance for postdoctoral work, and are work resources [12, 19]. The third category of life factors consists of time, psychological, and organizational demands at work, which are factors that require long-term commitment from postdocs and are job demands [4, 25]. The fourth category of individual perceptual factors has a direct influence on the psychological state and affective perception during postdoctoral work [24]. And in the JD-R model, these factors serve as potential channels through which job demands and job resources influence job satisfaction [26].

Existing studies have confirmed that elements of job characteristics are closely related to postdoctoral job satisfaction, indicating that the JD-R model has a strong explanatory capacity for postdoctoral job satisfaction. However, the positive and negative effects of job characteristic elements on job satisfaction have not been organically combined to form a balanced and comprehensive analytic framework with interaction. Based on this, the JD-R model is selected as the analytical perspective, and the postdoctoral job characteristics are analyzed from the two core dimensions of job demands and job resources, in order to deeply and comprehensively depict the influencing factors and the mechanism of postdoctoral job satisfaction, and to provide empirical evidence for comprehensively and deliberately enhancing the postdoctoral job satisfaction.

## Research hypothesis

In 1988, American psychologist Stephen Hobfoll first proposed the conservation of resource theory (COR), which asserts that individuals always seek and possess resources, subconsciously perceive resource loss as a threat, become psychologically stressed and tense, and engage in self-protection when they perceive that existing resources are depleted or desired resources are unavailable. When they perceive that existing resources are depleted or that desired resources are unavailable, they generate psychological stress and tension and engage in self-defense [27]. Resource conservation theory describes the interaction of resources between the individual and the social environment (Hobfoll, 1988, 1990), while emphasizing the role of individual resource factors in predicting job satisfaction from an evolutionary standpoint, i.e., individuals are motivated to acquire and protect their resources by the need to adapt to the environment and maintain survival [28]. This fundamental assumption is also central to explaining the evolution of human psychology and behavior, confirming the theory of organizational behavior that proposes job satisfaction is the result of congruence between individual and job characteristics and providing a theoretical foundation for the JD-R model [29]. The JD-R model proposed by Demerouti, Bakker, Nachreiner, and Schaufeli subsequently refined the job characteristic factors that affect employees' physical and mental health and work conditions into two categories: job demands and job resources. Job demands refer to the physical, psychological, social, or organizational demands of work, such as job stress, role burden, role conflict, and time pressure, which necessitate continuous effort and result in negative perceptions of the workplace [30]. These demands require sustained physical or mental effort or skill and are therefore associated with a certain level of physical and psychological exertion, which can result in negative perceptions of the job. Their operational indicators consist of emotional demands, interpersonal demands, workload, time strain, job responsibilities, role conflict, work-family conflict, and physical environment [31]. Alternatively, job resources are work factors that provide support and assistance to workers, such as the work environment, social support, wage compensation, and job security [32], these factors can contribute to the accomplishment of work objectives, reduce the physical and mental demands of the job, and stimulate personal growth, learning, and development [33]. and can motivate employees to work [30]. The operational indicators are: job control, social support, feedback, compensation,

career opportunities, task importance, supervision and guidance, and organizational justice [30, 34]. Schaufeli proposed the JD-R extension model, which includes positive correspondence, adds job engagement, and considers fatigue and job engagement to be mediators between job characteristics and job satisfaction [33, 35]. The JD-R model has thus far generated three central hypotheses. The first hypothesis is the "dual path" hypothesis, which states that there are two ways in which employment influences employees: the loss path and the gain path. The second hypothesis is the buffering hypothesis, which states that job resources can mitigate the negative effects of job demands by mitigating the attrition of employees with high job demands. The third hypothesis is the response hypothesis, which states that under elevated job demands, job resources are more likely to increase job engagement and motivation. Employees will be motivated to maximize their use of job resources in order to better engage in their work and achieve their work objectives when job demands are high [36]. On the basis of the JD-R model's dynamic cyclical framework for analyzing postdoctoral job satisfaction, the following hypotheses are proposed.

**Two-path hypothesis of the influence of job characteristics on postdoctoral job satisfaction.** Scholars have advanced the study of the JD-R model by testing the mode's hypotheses. Some of these studies have examined the two psychological processes underlying the model's hypotheses, revealing a direct predictive relationship between these two psychological processes and job satisfaction [37]. The first path is the path of attrition, in which individuals' positive perceptions are negatively impacted by job demands. The logic is that increasing job demands consume employees' resources, leading to a loss of resources, which may directly lead to health problems or negative perceptions of job satisfaction, forming the "attrition path" of job satisfaction [7]. The second path is the gain path, in which positive perceptions are positively influenced by job resources [38]. According to self-determination theory, the degree to which an individual self-perceives satisfaction is dependent on the degree to which their three fundamental requirements of competence, autonomy, and belongingness are met. And job resource orientation is closely related to both individual autonomy and belongingness satisfaction [39]. On the gain path, an increase in job resources such as performance feedback, salary and compensation, and autonomy in decision-making can directly satisfy the needs of employees' competence and autonomy, provide job security for individuals, promote their positive perception of job satisfaction, and aid them in achieving their job objectives effectively [40]. The dual-path hypothesis of the JD-R model of job demands and job resources and positive emotional perceptions (e.g. job satisfaction) has been more consistently supported by studies based on samples from various countries and occupations [32, 41–43].

In the postdoctoral job satisfaction study, existing studies have measured and confirmed the positive predictive effect of certain types of job resources on postdoctoral job satisfaction from micro perspectives such as supervisor support [22], organizational psychological support [44], and organizational system [45], respectively. For instance, researchers discovered that postdoctoral perceptions of organizational satisfaction were heavily influenced by the academic organization's evaluation, recruiting practices, research support, and training programmes [45]. Similarly, other researchers have discovered that certain types of job demands, such as time pressure, academic pressure, and work-life imbalance, are direct causes of decreased job satisfaction or even departure from academic careers among postdocs [15], which is consistent with the dual path hypothesis of the JD-R model. Based on this, research hypothesis H1 was proposed:

H1a: Job demands are negatively related to postdoctoral job satisfaction;

H1b: Job resources are positively related to postdoctoral job satisfaction.

**Hypothesis of the mediating effect of job burnout and job engagement on postdoctoral job satisfaction.** According to the original JD-R model, both job demands and job resources will have an independent effect on job satisfaction. As the model has been continuously used and improved, many scholars have begun to investigate the various manifestations of job demands and job resources in greater detail, while also paying greater attention to the specific paths of their roles [26]. In the two paths of job demands and job resources, the JD-R model identifies two potential psychological processes: the repression process of fatigue and the motivation process of motivation. The initial psychological process of exhaustion is repression. Job demands include the continuous physical and psychological efforts required in work, and when job demands are high, extra efforts must be made to achieve job goals; this process will produce physical and psychological costs, such as fatigue and irritability, which will deplete the individual's physical or mental energy if this state persists [7]. Consequently, high job demands can lead to physical and mental exhaustion in employees, which may result in health problems or psychological issues, which in turn reduces motivation to work and the phenomenon of withdrawal [46]. The second psychological process is the motivational process. Job engagement is a work-related, positive, and gratifying cognitive state that manifests as a highly pleasurable and motivating trait of the employee [47]. And work resources are embedded social, psychological, physiological, and organizational resources that can contribute to the achievement of job objectives, the reduction of job demands and associated physiological and psychological costs, the promotion of job engagement, and the enhancement of job satisfaction perceptions [32]. The academic subordination of postdocs has been found to necessitate that they simultaneously assume multiple roles and identities, and their cognitions and behaviours generate more readily and intensely conflicting pressures in response to high job demands [48], inhibiting their positive emotions at work and generating negative perceptions such as decreased interest in work and poor personal fulfillment, which in turn have negative effects on job satisfaction. In contrast, financial support and other salary resources can not only directly affect postdoctoral academic career development, but they can also indirectly affect postdocs by influencing their academic commitment and academic motivation [49], thereby becoming an endogenous factor that enhances postdoctoral job satisfaction [50]. Accordingly, research hypothesis H2 is proposed:

H2a: Job demands have a negative indirect effect on postdoctoral job satisfaction through elevated job burnout;

H2b: Job resources have a positive indirect effect on postdoctoral job satisfaction by enhancing job engagement.

**The interaction hypothesis of job demands and job resources on postdoctoral job satisfaction.** Although the JD-R model illustrates the independent effects of job demands and job resources on positive job perceptions, it also emphasizes the interaction between these two factors and their impact on job satisfaction [38]. On the one hand, job demands are more likely to be met in a more resourceful environment, which reduces the likelihood of job demands causing job stress and buffers the negative effects of high job demands on individuals, also known as the "buffering hypothesis"; on the other hand, in a more demanding environment, individuals are less likely to be inert, and are able to engage more fully in their work, mobilizing the large amount of work resources they already have to acquit the demands of the environment. This is also known as the "coping hypothesis" [37]. Three substantial investigations provide support for this hypothesis. In a study with a sample of teachers, Bakker, Hakanen, Demerouti, et al. found that the positive effect of high job resources on job performance was more

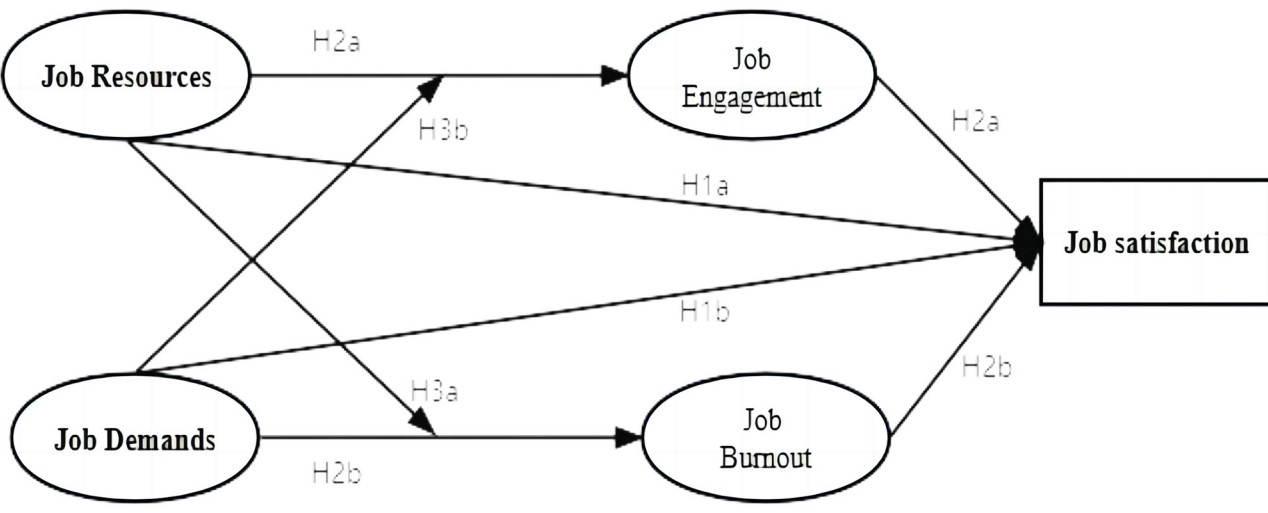

**Fig 1. Theoretical model of postdoctoral job satisfaction based on JD-R model.**

pronounced when teachers were in an environment with high job demands (e.g., high levels of disruptive student behavior) [41]. In a subsequent study of a sample of dentists, Hakanen, Schaufeli, and Ahola discovered that job resources had a notably significant impact on employees' ability to maintain high levels of engagement despite high job demands [51]. In another study, Bakker et al. examined the effects of the interaction of job resources and job demands on employees' task performance and organizational commitment. They discovered that high job resources had a positive impact on employees' task performance and organizational commitment when they were faced with high job demands. The study found that when employees are confronted with high job demands, high job resources have the greatest motivational influence on them, particularly when employees have the highest levels of organizational commitment and task performance [52]. Existing studies have found that postdocs face multiple stressors, such as short-term contracts, tenure changes, and high appraisal demands, and job resources, such as career development, professional growth, and work-life support, have substantial positive impacts on their psychological health [53]. Likewise, the positive effect of job resources on their psychological well-being increased substantially as job demands increased, and there was an interaction between the two on job satisfaction. Accordingly, research hypothesis H3 was proposed:

H3a: Job demands can enhance the positive impact of job resources on postdoctoral work engagement;

H3b: Job resources can reduce the positive effect of job demands on postdoctoral burnout.

Based on the above, we constructed a postdoctoral job satisfaction theoretical model (Fig 1).

## Research methods

### Data overview

In 2020, 7,670 postdocs from 93 countries worldwide participated in a global postdoctoral survey conducted by *Nature*, which provided the data for this study. In terms of subject areas, 51.9% of respondents were in biomedical sciences and only 4.7% were in social sciences; 38.6%

of respondents conducted their postdoctoral research in their home countries, 61.4% were not in their home countries, and 17.9% were at the same postdoctoral institution as their doctoral studies. In terms of gender and age structure, 3578 (47%) were male and 4001 (52%) were female; 2069 (27%) were under 30 years old; 5034 (66%) were between 31 and 40 years old; and 537 (7%) were under 40 years old. After excluding non-postdoctoral staff, part-time post-doctoral staff, missing values, and invalid samples, 5861 samples were finally screened and retained.

Before conducting a formal study, *Nature* considered potential cross-cultural regional disparities and utilized exploratory interviews to make sure the questionnaire's items were consistent among respondents from various cultural backgrounds. A considerable amount of reliability exists for the questionnaire. In this study, the appropriate questions from the respective portions of the questionnaire were chosen to describe the variables being examined.

**Measurement indicators.**

*1. Job satisfaction indicators.* The postdoctoral job satisfaction metric (question 43: How satisfied are you with your current postdoctoral work?) was the dependent variable. On a 7-point scale, "1" stood for "very dissatisfied" and "7" for "very satisfied," with "1" denoting "very dissatisfied" and "7" denoting "very satisfied."

*2. Job demand indicators.* The JD-R model and prior research indicated that job demands are concerned with the physical, social, or organizational elements of an employee's work that call for perseverance and hard work to complete, and typically include both objective indicators like the number of work tasks and the person's subjective perceptions of work tasks [54]. Due to this, this study defines job demands as time demands and life demands, based on the frequency of overtime and work-life balance.

*3. Job resource indicators.* Job resources are defined as physical, psychological, social, and organizational factors that can help individuals achieve work goals, cut down on work expenses, and promote personal growth in accordance with the JD-R model definition and measurements of work resources in previous studies [38]. Job resources can be further broken down into relational resources, which are embedded in interactions with coworkers; task resources, which are embedded in job autonomy; and organizational resources, which are embedded in career advancement [31]. Combining the classification of extant literature with the actual circumstances of postdoctoral work [6], a total of 11 indicators of salary and compensation, career promotion, job security, training and learning, work environment, cooperative mentoring, colleague relationships, organizational culture, organizational commitment, psychological support, and organizational culture were chosen as work resource indicators from three dimensions: organizational resources, relational resources, and humanistic resources. A confirmatory factor analysis of the three job resource dimensions outlined above yielded a model with a good fit (2/ df = 2. 998, RMSEA = 0. 055, CFI = 0. 913, TLI = 0. 898, NFI = 0. 875).

*4. Job burnout indicators.* Burnout is a symptom of excessive stress that includes mental exhaustion, depersonalization, adopting an indifferent and neglectful attitude toward work objects and the environment, and a negative evaluation of individual's work [55]. Academic burnout is an extension of job burnout, which is characterized by researchers' lack of enthusiasm for scientific tasks, skepticism regarding the value of research projects, and denial of their own research skills [56]. Friedman pointed out that one of the key indicators of job burnout is the decreased willingness to work caused by the gap between ideals and reality [57]. Based on this, the three variables of "anxiety and depression", "work expectation

**Table 1. Variable descriptions.**

| Variable | Specific variables | Variable Description | Cronbach's α |
|---|---|---|---|
| Job satisfaction | Postdoctoral job satisfaction | Fixed order, 1 = very dissatisfied, 7 = very satisfied | 0.879 |
| Job demands-Time demands | Frequency of overtime work | Fixed order, 1 = never work overtime, 6 = work overtime more than 20 times | 0.818 |
| Job demands-Life demands | Work-life balance | Constant order, 1 = perfectly balanced, 7 = perfectly unbalanced | 0.847 |
| Job resources-Organizational Resources | Wages and salaries | Fixed order, 1 = very dissatisfied, 7 = very satisfied | 0.910 |
| | Career advancement | Fixed order, 1 = very dissatisfied, 7 = very satisfied | 0.767 |
| | Job security | Fixed order, 1 = very dissatisfied, 7 = very satisfied | 0.730 |
| | Training and learning | Fixed order, 1 = very dissatisfied, 7 = very satisfied | 0.808 |
| | Working environment | Fixed order, 1 = very dissatisfied, 7 = very satisfied | 0.879 |
| Job resources-Relational-resources | Co-mentorship | Fixed order, 1 = very dissatisfied, 7 = very satisfied | 0.843 |
| | collegiality | Fixed order, 1 = very dissatisfied, 7 = very satisfied | 0.767 |
| Job Resources-Humanistic resources | Organizational culture | Fixed order, 1 = very dissatisfied, 7 = very satisfied | 0.876 |
| | Organizational commitment | Fixed order, 1 = very dissatisfied, 7 = very satisfied | 0.793 |
| | Organizational Recognition | Fixed order, 1 = very dissatisfied, 7 = very satisfied | 0.827 |
| | Psychological Support | Fixed order, 1 = strongly disagree, 7 = strongly agree | 0.891 |
| Job Burnout | Anxious depressive state | Definite order, 1 = not present, 2 = uncertain, 3 = anxiety depression present | 0.916 |
| | Work expectation deviation | Fixed order, 1 = better than I thought, 2 = meets expectations, 3 = worse than I expected | 0.879 |
| | Willingness to leave | Definitive order, 1 = does not plan to leave current postdoctoral position in the next 6–12 months, 2 = not sure, 3 = does plan to leave | 0.783 |
| Job Engagement | Personal fulfillment | Fixed order, 1 = very dissatisfied, 7 = very satisfied | 0.767 |
| | Work Interest | Fixed order, 1 = very dissatisfied, 7 = very satisfied | 0.812 |

deviation" and "willingness to leave" were chosen as the postdoctoral burnout indicators for this study.

5. *Job engagement indicators.* Job engagement is the amount of physical, cognitive, and emotional energy individuals invest in their occupation [58]. According to the JD-R model, both time devoted to work and being enthusiastic about work reflect the degree of work engagement of individuals. The Job Engagement Scale created by Yewen Zhang and Yiqun Gan measures three dimensions of job engagement: vitality, commitment, and concentration. Vitality refers to the high energy level in work, which is the ability to be energetic, adaptable, willing to devote energy to work; commitment refers to the process of devoting oneself to work, which is accompanied by enthusiasm and meaning and brings a sense of pride and inspiration; concentration refers to the identification with work and the capacity to experience the meaning and challenge of work [59]. Based on this, "personal fulfillment" and "work interest" were selected as indicators of postdoctoral job engagement for measuring the postdoctoral work status. Variable descriptions are shown in Table 1. The Cronbach's alpha coefficients for each variable reached the acceptable level of 0.7 suggested by the study, indicating good reliability [60].

## Statistical methods

SPSS 27.0 and AMOS 26.0 was used for statistical analysis in the study. The statistical analysis consisted of the following steps: First, descriptive statistical analysis was used to describe the

sample's fundamental characteristics, such as mean, standard deviation, and variance; then, confirmatory factor analysis was used to assess the discriminant validity of the model. Second, the dual-path theoretical hypothesis (H1) was tested by analyzing the means and correlation coefficients of each variable; third, the mediating roles of job engagement and burnout in the dual path were examined by regression analyses (H2); Third, hierarchical regression analyses were conducted to examine the moderating roles of job resources and job demands on the dual-path hypothesis; among them, the regression coefficients are explored by maximum likelihood estimation in the two-path test; and the self-help method (Bootstrap Method) is utilized in the tests of the mediating and moderating effects, and the self-help method is a more effective and powerful method of testing the effects of mediating variables, and it is more effective and powerful in testing the effects of indirect variables. It makes no assumptions about the sampling distribution of indirect effects and can be used to infer indirect effects in any model of the intermediate variables [61].

## Results

### Descriptive statistical analysis

The study begins with a descriptive statistical analysis using SPSS 27.0 of the principal latent variables (Table 2). The analysis revealed that the mean level of job satisfaction among postdocs in the sample was 4.57, which was greater than the median level of 4, indicating that the average level of job satisfaction among postdocs in the sample was high. On the dimension of job resources, the mean values of organizational resources, relational resources, and humanistic resources were higher than 4, indicating that the postdoctoral group had a more positive perception of job resources. The mean value of relational resources is 4.83, which is more than 0.5 higher than the mean values of organizational resources and humanistic resources. Relational resources are the most prominent aspect of work resources as perceived by postdocs. In the job demand dimension, the mean values of time demand and life demand are 1.86 and 3.42, respectively, which are less than the median value of 4, indicating that the postdoctoral group in the sample has a limited perception of job demands overall. Two dimensions, personal fulfillment and job interest, had mean values of 4.58 and 5.45, while three dimensions,

**Table 2. Descriptive statistics.**

| | N | Minimal value | Maximum value | Average value | | Standard deviation | Variance |
|---|---|---|---|---|---|---|---|
| | Statistical quantities | Statistical quantities | Statistical quantities | Statistical quantities | Standard error | Statistical quantities | Statistical quantities |
| Organizational resources | 5861 | 1.00 | 7.00 | 4.1015 | .01565 | 1.19798 | 1.435 |
| Relational resources | 5861 | 1.00 | 7.00 | 4.8270 | .01868 | 1.43030 | 2.046 |
| Humanistic resources | 5861 | 1.00 | 7.00 | 4.2037 | .01576 | 1.20621 | 1.455 |
| Time demands | 5861 | 1.00 | 6.00 | 1.8608 | .01860 | 1.42422 | 2.028 |
| Life demands | 5861 | 1.00 | 7.00 | 3.4213 | .02248 | 1.72083 | 2.961 |
| Personal fulfillment | 5860 | 1.00 | 7.00 | 4.5800 | .02201 | 1.68523 | 2.840 |
| Work interest | 5860 | 1.00 | 7.00 | 5.4546 | .01963 | 1.50240 | 2.257 |
| Anxious depressive state | 5861 | 1.00 | 3.00 | 1.7558 | .01060 | .81156 | .659 |
| Willingness to leave | 5861 | 1.00 | 3.00 | 1.9800 | .01181 | .90379 | .817 |
| Work expectation deviation | 5861 | 1.00 | 3.00 | 2.1977 | .00837 | .64047 | .410 |
| Job satisfaction | 5854 | 1.00 | 7.00 | 4.5743 | .02122 | 1.62319 | 2.635 |
| Effective N | 5852 | | | | | | |

**Table 3. VIF inspection.**

|  | Organizational resources | Relationalresources | Humanistic resources | Time demands | Life demands | Personal fulfillment | Work interest | Anxiou depressive state | Willingness to leave | Work expectation deviation |
|---|---|---|---|---|---|---|---|---|---|---|
| Tolerance | 0.528 | 0.552 | 0.459 | 0.943 | 0.860 | 0.463 | 0.503 | 0.927 | 0.887 | 0.740 |
| VIF | 1.893 | 1.812 | 2.180 | 1.060 | 1.163 | 2.160 | 1.986 | 1.079 | 1.127 | 1.352 |

anxiety and depression state, willingness to leave, and job expectation deviation, had mean values of 1.76, 1.98, and 2.20, indicating that both job engagement and job burnout were at a moderately high level.

**VIF inspection.** Before the regression analysis of the thesis data, it is necessary to test the data for the existence of multicollinearity using VIF, and the results are shown in Table 3. The maximum VIF between the variables is 2.180, which is less than 10, and the minimum tolerance is 0.459, which is greater than 0.1, so it can be seen that there is no serious problem of multicollinearity between the variables [62].

**Confirmatory factor analysis for discriminant validity.** Meanwhile, we examined the dependability of the selected survey data to assess the consistency, dependability, and stability of the characterization of the selected items. The results showed that the Cronbach's alpha coefficients for each variable surpassed the standard of 0.7 [63, 64], suggesting good internal consistency reliability. Also, based on the methodology of Netemeyer, Johnston, Burton, and Wang et al. [65, 66], we used the dimensions as latent variable indicators and applied validated factor analysis to test the variable discriminant validity. As can be seen from Table 4, the 4-factor model ($X^2$ = 467.77, df = 179, $X^2$/df = 2.613, CFI = 0.963, RMSEA = 0.058, TLI = 0.943) fits better than all the other models and has a good matching index, which implies that there is very good discriminative validity among the four variables in this study.

Following the recommendations of Wang et al, Fornell and Larcker [67] and Netemeyer, Johnston & Burton [65], we further examined the convergent validity of the four dimensions using average extracted variance (AVE). The AVE values for each dimension ranged from 0.60 to 0.67, which were above the critical value of 0.5 [64], indicating good convergent validity [65]. Combining the results of CFA and the above analyses, it can be proved that the four variables have good discriminant validity, so the next analysis can be carried out.

## Research hypothesis testing

**A two-path hypothesis regarding the relationship between job characteristics and postdoctoral job satisfaction.** According to the means and correlation coefficients of the variables depicted in Table 5, it is evident that job demands are all significantly negatively

**Table 4. Confirmatory factor analysis results.**

|  | $X^2$ | df | $X^2$/ df | CFI | RMSEA | TLI |
|---|---|---|---|---|---|---|
| four-factor model | 467.77 | 179 | 2.613 | 0.963 | 0.058 | 0.943 |
| three-factor model | 693.21 | 167 | 4.151 | 0.820 | 0.099 | 0.900 |
| two-factor model | 1752.69 | 167 | 10.495 | 0.830 | 0.179 | 0.810 |
| single-factor model | 6609.13 | 170 | 38.877 | 0.760 | 0.358 | 0.730. |

Note: The four-variable model contains four variables: job resources, job demands, job engagement, job burnout; the three-variable model contains three variables: job resources + job demands, job engagement, job burnout; the two-variable model contains two variables: job resources + job demands, job engagement + job burnout; and the univariable model contains one variable: job resources + job demands + job engagement + job burnout. The "+" stands for combining into one variable.

**Table 5. Means and correlation coefficients for each variable.**

|  | Organizational Resources | Relational resources | Humanistic resources | Time Demands | Life Demands | Job Resources | Job Demands | Job Satisfaction |
|---|---|---|---|---|---|---|---|---|
| Organizational Resources | 1 | | | | | | | |
| Relational Resources | .496*** | 1 | | | | | | |
| Humanistic resources | .641*** | .583*** | 1 | | | | | |
| Time Demands | -.170*** | -115*** | -.139*** | 1 | | | | |
| Life Demands | -.309*** | -.221*** | -.298*** | .190*** | 1 | | | |
| Job Resources | .881*** | .731*** | .901*** | -.170*** | -334*** | 1 | | |
| Job Demands | -318*** | -.223*** | -.292*** | .719*** | .818*** | -.336*** | 1 | |
| Job Satisfaction | .569*** | .602*** | .570*** | -.137*** | -.296*** | .669*** | -.290*** | 1 |
| Average value | 4.1015 | 4.8270 | 4.2037 | 1.8608 | 3.4213 | 4.2650 | 2.6410 | 4.5743 |
| Variance | 1.435 | 2.046 | 1.455 | 2.028 | 2.961 | 1.138 | 1.480 | 2.635 |

Note:

**** $p < 0.001$

associated with job satisfaction, whereas job resources are all significantly positively associated with job satisfaction. This indicates that postdoctoral job satisfaction decreases as job demands increase. In contrast, an increase in postdoctoral possession of job resources may initiate a motivational process that enhances their perceptions of job satisfaction. Thus, both the attrition path hypothesis and the gain path hypothesis are substantiated.

**Hypothesis of the mediating effect of job burnout and job engagement on postdoctoral job satisfaction.** According to the steps of mediation effect testing [61], this paper uses the Bootstrap method to test the mediation path hypothesis of job demands by triggering job burnout and hindering satisfaction perception and the two mediation roles of job resources by stimulating job engagement and improving job satisfaction in the JD-R model (Hypothesis H2). The number of repetitions of this sample is 5000, and the confidence interval is set to 95% [68]. According to Table 6's estimation results, the effect ratio of job engagement on job resources is 77.480%, and the effect ratio on the three sub-dimensions of job resources, organizational resources, relational resources, and humanistic resources, is 76.751%, 84.752%, and

**Table 6. Standardized bootstrap mediation effects test results.**

| Paths | Intermediary Efficacy value | Total effect value | standard error | 95% confidence interval | efficiency ratio |
|---|---|---|---|---|---|
| Job resources = > Job engagement = > Job satisfaction | 0.066** | 0.085** | 0.001 | 0.503 ~ 0.534 | 77.480% |
| Organizational resources = > Job engagement = > Job satisfaction | 0.118** | 0.154** | 0.003 | 0.420 ~ 0.452 | 76.751% |
| Relational resources = > Job engagement = > Job satisfaction | 0.289** | 0.342** | 0.006 | 0.493 ~ 0.526 | 84.752% |
| Humanistic resources = > Job engagement = > Job satisfaction | 0.131** | 0.153** | 0.003 | 0.469 ~ 0.502 | 85.150% |
| Job demands = > job burnout = > job satisfaction | -0.073** | -0.193** | 0.008 | -0.121 ~ -0.097 | 37.679% |
| Time demands = > job burnout = > job satisfaction | -0.068** | -0.157** | 0.007 | -0.072 ~ -0.047 | 43.134% |
| Life demands = > Job burnout = > Job satisfaction | -0.102** | -0.279** | 0.009 | -0.121 ~ -0.096 | 36.586% |

Note:

** $p < 0.01$

85.150%, respectively, and none of the 95% confidence intervals of the mediating effects calculated by the self-help method contain 0, i.e., the effect ratio of job engagement on both job resources and the all three sub-dimensions of work resources are partially mediated, so the H2a hypothesis is confirmed. The percentage of mediating effect of job burnout on job demands is 37.679%, and the percentage of effect on two sub-dimensions of job demands, time demands and life demands, is 43.134% and 36.586%, and none of the 95% confidence intervals of mediating effects calculated by the self-help method contains 0. It proves that there is a partially mediating effect of job burnout on the relationship between time demands, life demands, and job satisfaction among postdocs. The H2b hypothesis was confirmed. That is, job resources have a positive indirect effect on job satisfaction through job engagement, and job demands have a negative indirect effect on job satisfaction through job burnout. Thus, the hypothesis H2 is supported.

**The interaction hypothesis of job demands and job resources on postdoctoral job satisfaction.** Referring to the existing literature [69], this paper tests hypothesis3 by constructing Eqs (1)-(2),which demonstrate that the indirect effect of job demands/resources on postdoctoral job satisfaction (through job engagement/job burnout) is relatively small in the high level group of job resources/demands, but relatively large in the low level group. Here is the specific test equation:

$$M = \alpha_0 + \alpha_1 X + \alpha_2 W + \alpha_3 X \times W \tag{1}$$

$$Y = \beta_0 + \beta_1 M + \beta_2 X + \beta_3 W + \beta_4 X \times W \tag{2}$$

X represents the independent variable (job demands/resources), M represents the mediating variable (job burnout/job engagement), W represents the moderating variable (resources/demands), and Y represents the dependent variable (job satisfaction). Substituting Eq (1) for Eq (2) yields Eq (3):

$$Y = (\beta_0 + \alpha_0\beta_1) + [\beta_2 + \beta_1(\alpha_1 + \alpha_3 W)] \times X + (\beta_3 + \beta_1\alpha_2)W + \beta_4 X \times W \tag{3}$$

The conditional indirect effect magnitude of the mediating role in Eq (3) is $\beta_1 (\alpha_1 + \alpha_3 W)$ [70]. In order to reduce multicollinearity, the independent and moderating variables were first centered and grouped according to the moderating variables according to (M ± 1 std) [71], if the difference in the conditional indirect effect size of the mediated role in both cases does not contain 0 at the 95% confidence interval when the moderating variable W is considered to be high (1 standard deviation above the mean) or low (1 standard deviation below the mean), respectively, the mediated role of the moderated role can be assumed to be supported [68]. Table 7 displays the estimation results of the self-help method test for the moderated-mediated effect. According to Table 7, hypotheses H3a and H3b are supported on the life demands dimension and the organizational and humanistic resource dimensions, respectively. The difference in the indirect effect of job resources on individual job satisfaction through the mediating variable of job engagement is not statistically significant in the high time demand cohort relative to the low time demand cohort (the 95% confidence interval of the difference between the indirect effect groups contains 0), thus H3a is not supported in the time demand dimension. Likewise, H3b was not supported on the dimension of relational resources, and the H3 hypothesis was only partially examined.

## Discussion

Using the JD-R model as the theoretical analysis framework and the survey data of 2020 Nature global postdoctoral survey, this study explored the relationship between job

**Table 7. Moderated mediation model test results.**

| | | Effect size factor | Standard error | Lower limit | Upper limit |
|---|---|---|---|---|---|
| Job resources—Job engagement—Job satisfaction | Low time demands | 0.067 | 0.002 | 0.064 | 0.070 |
| | High time demands | 0.066 | 0.002 | 0.063 | 0.070 |
| | Between-group variability (low-high) | -0.001 | 0.002 | -0.004 | 0.002 |
| | Low life demands | 0.059 | 0.002 | 0.055 | 0.063 |
| | High life demands | 0.066 | 0.002 | 0.063 | 0.070 |
| | Between-group variability (low-high) | 0.007 | 0.002 | 0.003 | 0.011 |
| Job demands—Job burnout—Job satisfaction | Low organizational resources | -0.049 | 0.007 | -0.062 | -0.036 |
| | High Organizational Resources | -0.030 | 0.008 | -0.045 | -0.014 |
| | Between-group variability (low-high) | 0.010 | 0.004 | 0.001 | 0.019 |
| | Low relational resources | -0.057 | 0.007 | -0.071 | -0.044 |
| | High Relational resources | -0.041 | 0.008 | -0.057 | -0.026 |
| | Between-group variability (low-high) | 0.016 | 0.010 | -0.003 | 0.034 |
| | Low Humanistic resources | -0.047 | 0.007 | -0.059 | -0.035 |
| | High Humanistic resources | -0.024 | 0.007 | -0.038 | -0.009 |
| | Between-group variability (low-high) | 0.023 | 0.009 | 0.006 | 0.040 |

characteristics elements and postdoc job satisfaction and their mechanisms of action, and primarily obtained the following findings.

## Job demands and job resources negatively and positively predict postdoctoral job satisfaction respectively

First, both types of job demands negatively affected postdoctoral self-perceived job satisfaction levels, and the JD-R model's attrition path hypothesis that job demands negatively predict postdoctoral job satisfaction was supported. Consistent with other socioeconomic and cultural contexts or occupational samples [31]. This suggests that excessive work demands may trigger negative personal job emotions, which may diminish postdoctoral self-perceptions of job satisfaction. Consistent with the findings of previous studies, postdocs, who are typically neither instructors nor students, are frequently mired in a rut and receive neither recognition nor the benefits they believe they deserve (e.g., family health insurance) [71, 72]. According to previous research, postdocs frequently experience uncertainty and insecurity due to the transient nature of their employment, the high expectations of their employers, and the intensely competitive job market [73]. This situation is exacerbated by increased administrative responsibilities and decreased research time [74–76], as well as by competition for positions and employment opportunities [76, 77], which results in the intrusion of work into postdoctoral life [74, 75]. It is difficult to separate the roles of work and life [78] and these conflicts are the primary cause of low job satisfaction among postdoctoral researchers. The JD-R model's hypothesis regarding the gain path of job resources positively predicting postdoctoral job satisfaction was supported by the findings of this study. This is consistent with previous findings in the literature, suggesting that postdoctoral training organization resource support for young scholars is an effective facilitator that plays a positive role in the development of job satisfaction among young scholars. This affirms previous theories about resource support as a mechanism for facilitating the academic career identity of young scholars [53], and that support from the organization can have a significant impact on the work ethic of the postdoctoral group, thereby enhancing the identity of academic research [79]. The findings support the resource conservation theory's hypothesis that resources are a central element in the process of job stress

generation and coping, and that individuals who lack resources will struggle to cope with stress and generate job satisfaction [80]. Moreover, the findings suggest that the postdoctoral group has a strong need for learning, social, and professional development and that adequate organizational, interpersonal, and humanistic resources can meet the fundamental needs of postdocs for competence development, autonomy, and organizational belonging, motivating individuals to pursue growth and development and to experience more positive emotions such as job satisfaction [13].

## Burnout and work engagement play a mediating role in the dual path

It was found that, while job demands and job resources have a certain direct effect on postdoctoral job satisfaction, they cannot completely and directly reduce or enhance postdoctoral job satisfaction, and the emotional perception and behavioral tendency, i.e., job burnout and job commitment, embedded behind the two, are the intermediary factors of attrition and cultivation of postdoctoral job satisfaction. High job demands trigger postdoctoral feelings of powerlessness and exhaustion in the work process, which leads to unfavorable impressions of work, whereas job resources promote postdoctoral engagement with their work by providing job facilitation, which leads to higher job satisfaction [81]. This result is consistent with Lazarus and Folkman's theory of stress and coping, which states that stress is a transactional process between an individual and his environment and that when an individual perceives that environmental demands exceed his ability to meet those demands, he instinctively initiates an avoidance response in order to minimize losses [82]. The postdoctoral imbalance between high effort and low reward appears to elicit negative sentiments that contribute to a decline in job satisfaction. Thus, we confirm previous findings regarding the direct relationship between job stress and job satisfaction among academics, as well as their propensity to depart the profession [83, 84]. However, in the attrition pathway where job demands predict postdoctoral job satisfaction negatively, job demands are not intrinsically a negative stressor and only become a negative stressor when a significant number of ongoing job demands prohibit postdocs from recovering effectively [41]. Similarly, in the gain path, the various material, psychological, and interpersonal resources obtained at work serve to alleviate postdoctoral negative emotions, encourage postdocs to participate actively in research, and directly increase postdoctoral satisfaction. On the other hand, it translates into job engagement, including self-esteem and a sense of accomplishment while completing tasks. When postdocs perceive or evaluate the external support environment positively, they are more willing to improve their professional knowledge and research skills by participating in research activities, and this willingness to participate actively becomes an endogenous factor influencing postdoctoral research engagement [50]. This is consistent with the "acquisition spiral" in resource conservation theory, which states that individual engagement can not only counteract resource loss but also stimulate the generation of new resources, so that resource engagement can generate "compound interest." This is consistent with the "acquisition spiral" in the theory of resource conservation, which states that individual engagements not only counteract depletion but may also activate new resources, resulting in a "compounding" effect of resource engagements [28]. The findings also support the broaden-and-build theory of positive emotions, which links positive emotions to adaptive coping strategies in a positive manner [85].

## Job resources can partially buffer the risk of job burnout, and job demands can partially enhance the sense of job engagement

The buffering effect of job resources can assist postdocs in coping with stress induced by outside threats and reduce the risk of burnout due to job demands. Reaffirming the JD-R model's

setting that job resources can mitigate the negative effects of job demands on job satisfaction along the path of attrition [53], Previous research has demonstrated that supportive factors at the organizational levels enhance postdoctoral ability to meet high job demands and motivate them to develop academic career enthusiasm, thereby preventing burnout [86]. According to the view of protective factors, individuals are more self-sufficient, more resilient for job tasks, and feel more challenged and accomplished by jobs when they have access to sufficient job resources [35, 42, 52]. They are able to adapt to changing job demands and effectively manage stressful situations [87]. For postdocs, identification with and love of academic careers are extremely important individual protective factors that can buffer or offset different types of stress, frustration, conflict, and other risk states [88], enhance postdoctoral resilience to work stress, and thus effectively prevent academic burnout. This study discovered, however, that not all job resources can mitigate the negative effects of job demands, and only organizational and humanistic resources can reduce the likelihood of job demands causing postdoctoral job stress. This suggests that for postdocs, resourcing factors such as support and motivation in the work-place can effectively mitigate burnout in the face of high job demands and therefore mitigate the negative effects of burnout on job satisfaction [89]. In addition, it was discovered that life demands can motivate postdocs to work intrinsically and individuals to actively utilize resources for self-actualization. This is consistent with the view of resource conservation theory that resources are valuable to employees, and preserving and obtaining resources is the main motivation for individuals to achieve their expected goals. Therefore, employees will be proactive in performing work tasks, striving to obtain and preserve resources, and even some employees may be inclined to take risks [87]. Low job demands can foster inertia, whereas high life demands can motivate individuals to achieve their job goals and use their job resources more efficiently and effectively. As a result of this "coping" effect, postdocs are motivated to fully enjoy the satisfaction that comes from their work and family duties, which accelerates individual growth and development and boosts job satisfaction [90]. This process expands the motivating effect of high job resources, which on the one hand motivates individuals to make full use of existing job resources to better accomplish their work tasks and directly improve postdoctoral satisfaction through their work; on the other hand, job demands are also transformed from passive demands to active internal motivation, which naturally enhances the positive influence of job resources on positive emotions such as job satisfaction. Lower job demands, on the other hand, may lead to lethargy and a lack of goals in postdocs [91]. However, in this study, the path of "coping" with job demands was rejected in the dimension of time demands, which might be due to postdoctoral working hours have exceeded normal working hours [92], and most of their time is spent on "extra" work, such as teaching, administration, and organizing meetings, etc., resulting in a lack of sufficient time for scientific research, which is the most important source of stress, affecting their perception of their roles [93], thus making it difficult to stimulate the motivation of postdocs.

## Conclusions

Based on survey data from the 2020 *Nature* Global Postdoctoral Survey, using a combination of descriptive statistics, correlation analysis and regression analysis, this study investigates the influence of job characteristic elements on postdoctoral job satisfaction within the theoretical framework of the JD-R model and finds that the attrition path of job demands negatively influencing postdoctoral job satisfaction and the gain path of job resources positively influencing postdoctoral job satisfaction are both supported, and job burnout and job engagement play a significant role. The buffer hypothesis and the correspondence hypothesis, which propose that job demands and job resources interact and jointly affect postdoctoral job satisfaction, were

partially supported. In the buffer hypothesis, only two types of job resources, organizational resources and humanistic feedback, can reduce the negative effect of job demands on postdoctoral burnout; in the alternative hypothesis, only life demands can enhance the positive effect of job resources on postdoctoral job engagement. The study's main contribution is evident in two aspects: first, it expands the understanding of the elements that influence job satisfaction in postdocs. The current research on postdoctoral job satisfaction is still in its infancy, and there is a dearth of systematic research on job characteristic aspects. As a result, this study organically integrates the positive and negative effects of job characteristic factors on job satisfaction from an empirical standpoint, argues the relationship between job resources, job demands, and postdoctoral job satisfaction, and delves deeper into the intrinsic role mechanisms of their interactions to form a balanced and comprehensive analytical framework with interactivity. Second, the study generally validates the JD-R model's applicability in the postdoc group and broadens its application reach.

The implications of the study for enhancing postdoctoral job satisfaction by enhancing organizational management are as follows: Firstly, managers should ensure "breakthrough" and "complementary" working hours, while reducing "exhaustive" and "low-maintenance" working hours, which are unrelated to actual work, and reasonably arrange postdoctoral tasks according to the principles of professionalism and high efficiency, so as to reduce the negative emotion perceptions brought about by excessive work demands [94]. Secondly, managers should provide postdocs with more adequate and higher-level resources that can meet the needs of individual development and enhance postdoctoral job satisfaction, particularly the resources required for postdoctoral professional development and psychological health, in order to assist postdocs in coping with job demands and mitigating job burnout caused by job demands, which can indirectly increase postdoctoral job satisfaction. For instance, allowing and encouraging postdocs to offer their own perspectives on their own duties or institutional work plans empowers them to feel in charge of their work [95]. Specifically, they can establish research teams to communicate regularly about work tasks and create a supportive working philosophy and research environment, as well as establish work incentive mechanisms to stimulate and enhance postdoctoral work efficiency and commitment by means of job promotion, performance allowance, paid leave, etc. In conclusion, postdoctoral stations must find a method to support postdocs that strikes a balance between efficiency and value, as well as consider the realistic needs of various postdoc groups, in order to increase postdoctoral satisfaction with their academic work.

This study has some limitations that require further investigation. First, because the Global Postdoctoral Survey encompasses 93 countries worldwide, the coverage is extensive, the regions are dispersed, and the study lacks control variables for regional differences. Second, the study determined the measurement indicators after considering the rationality of indicator measurement and the availability of data; however, the comprehensiveness of the indicator measurement may be limited by the rich conceptual connotation of the relevant variables. The next investigation may contemplate a breakthrough from two perspectives. On the one hand, the inclusion of corresponding individual variables (e.g., nationality, gender, etc.) can be considered to explore in depth the applicability of the job trait model to individuals with different characteristics, to explore the primary job demands that cause negative emotions in individuals with a certain trait, and also to identify the most effective job resources for individuals with a certain trait.

## Supporting information

**S1 Data.**
(XLSX)

## Acknowledgments

We acknowledge *Nature* in collecting and making available the data used in the research and acknowledge all the participants.

## Author Contributions

**Supervision:** Xinxing Duan.

**Writing – original draft:** Yue Zhang.

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
