## [Decision Letter · Decision Letter 0]

10 Jul 2023

PONE-D-23-15985Job Demands，Job Resources and Postdoctoral Job Satisfaction：An Empirical Study Based on the Data from Nature’s Global Postdoc SurveyPLOS ONE

Dear Dr. Duan,

Thank you for submitting your manuscript to PLOS ONE. After careful consideration, we feel that it has merit but does not fully meet PLOS ONE’s publication criteria as it currently stands. Therefore, we invite you to submit a revised version of the manuscript that addresses the points raised during the review process.

**Reviewer 1**. This paper examines the Job Demands,Job Resources, and Postdoctoral Job Satisfaction: An Empirical Study Based on the Data from Nature’s Global Postdoc Survey. The paper is in general not satisfactorily written and lacks novelty and justification for the need of study. I would like to make a few suggestions to improve the contributions of this research.

Abstract

The abstract should reflect the snapshot of the paper. It should be more persuasive. The Abstract lacks originality. Revisit the abstract specifically focusing on the following steps; (Brief glimpse into the topic, Purpose, Methodology, and Practical Implications).

Introduction

The Mechanism of introduction is not appropriate. It must include the territory of the study, niche, and detail about the current research, leading toward the need for this research. It lacks justification for; the need for the study, the selection of the variables, and the relevance of the population/context of the study. Furthermore, the introduction lacks to justify the contribution of this study to Academia and/or Industry, why is it worth studying? What`s new in it and why does it matter? In short, the entire introduction Part of the chapter needs serious attention.

Literature Review

Under the heading "1. Two-path hypothesis of the influence of job characteristics on post-doctoral job satisfaction” It is mentioned that earlier, studies have been conducted on the same topic in the same context, then the question arises, “What the current study is going to add in it??

A few references are NOT in APA style, as per the 7th edition. See, For example, on page 10, the last line of the second paragraph.

What do you mean by this term “In the university field? at page 10

Hypothesis H4 is contradictory to Hypothesis H2. H2 states that job demands have a negative indirect effect on postdoctoral job satisfaction……. While H4 states that “Postdoctoral job satisfaction is highest with the combination of “high demands-high resources”. Means positively associated…?? Kindly elaborate on how??

This section contains many grammatical mistakes and typo errors. Kindly proofread.

The scholar has failed to develop a link between IVs and DV. A number of claims require referencing. Hence, the Literature requires significant rework.

Research Methodology

Data collection methods, inclusion-exclusion criteria, Justification for sample size, etc. are not properly discussed.

It is mentioned that “The right questions from the respective portions of the questionnaire were chosen to describe the variables being looked at". HOW??? Is any specific method applied?? Discuss its justification as well.

No explanation of instrumentation is mentioned. E.g. job satisfaction questionnaire is adopted/adapted from?? Or was either developed??

Statistical Methods and Results

Why SPSS is used?? And if SPSS, then why the quite older version is used since we have many revisions after SPSS 22.0? Instead, it is recommended to use SEM and SMART-PLS.

Why two-factor ANOVA is used for mediation??? Can you provide any logical justification for it?

“Where the regression coefficients are explored in a two-path test with maximum likelihood estimation; and in the test of mediating and moderating effects using the self-help method (Bootstrap Method), What is this? Where is the moderator in the entire study?? How mediation is checked through the self-help method? Etc.

If SPSS, is used, then why not “Baron and Kenny” OR “Andrew Hayes approach” applied for mediation?

“The interaction hypothesis of job demands and job resources on postdoctoral job satisfaction” What is the need for all this?? Elaborate on the justification of its relevance.

“Table 7 Self-help method test for the mediated role of the moderated” What does this table indicate???

What about the normality of data? Any test applied?? discuss

The entire section does not have a single reference. The author is supposed to support its results from existing literature.

In short, this section requires some serious attention.

Discussion and Conclusion

Discussion should relate or contradict the study finding in light of existing research furthermore why your study found such results; there should be a discussion on results rather than just confirming the results.

The conclusion of the study should be drafted in light of the objectives of the study. The author should highlight how each of the objectives identified in the research was attained.

Final Evaluation Statement:

The paper contains serious issues regarding, the novelty of the paper, justification of the study, methodology applied, and analysis section, besides, formatting issues like language, sequence, spelling, grammar, and coherence).

**Reviewer 2**Introduction:

Introduction section focuses primarily on presenting facts and figures concerning the previous studies. It lacks the originality, significance, and contribution of your study. Why your conceptual framework is important to be studied, and how well it articulates the overall purpose of your study? The authors should highlight the role of job demands, resources in affecting the job satisfaction and why it is critical to examine motivational and hindrance mechanism to have deeper insights about the topic. Meanwhile, synchronize your framework with the overarching theory (i.e. JD-R) comprising the conceptual base of your study.

Literature Review:

Literature review section lacks clarity, coherence and relevance. For example, line 7 -12 of first para is supposed to outline two dimensions, however there are so many multi-factors have been highlighted without any due clarity of what your study aims to examine? It seems lack of understanding and objectivity. Likewise, putting citation (i.e. Weiss, Dawis, & England, 1967) in the end doesn’t sound good as it shows your study aims pointing to someone else work.

In second para, you stated that “After controlling for demographic variables, the research on the influencing factors of postdoctoral job satisfaction is primarily focused on a single dimension…………” Which research you are referring about? Please cite it with substantive evidence to support your claim.

In same para, you stated that “Second, the study lacks a better theoretical framework for evaluating and reflecting on postdoctoral job satisfaction from a systematic standpoint…………” which study lacks better theoretical framework? You need to revisit your literature review section it lacks substantive arguments and also have some technical writing issues.

Research Hypothesis:

In hypothesis section, you stated “The first hypothesis is the “dual path” hypothesis, which states that “there are two ways in which employment influences employees: the loss path and the gain path” however, it doesn’t make a sense to test the direct effect of job demands and resources on job satisfaction and call it the loss and gain path. Loss and gain paths were originally formulated to test outcome variables through job engagement and burnout, however testing direct paths to job satisfaction overrule such assumption. Revisit your statement and arguments.

In similar way, your second hypothesis uncovers …….“the buffering hypothesis, which states that job resources can mitigate the negative effects of job demands by mitigating the attrition of employees with high job demands”…….. however, in hypothesis section 2, you hypothesized the mediating role of burnout and engagement. Again it is misleading and vague. Revisit

Hypothesis section 3 aims to examine interaction effect of job resources and demands on job satisfaction. How comes the interaction effect would be carried out on the job satisfaction? Moreover, hypothesis H3a hypothesizes…… “job demands can enhance the positive impact of job resources on postdoctoral work engagement” …… How job demands can enhance this relationship? You haven’t provided any valid justification to this claim. Explain it

Hypothesis 4 is misleading too. How high demand-high resource combination produces the optimum satisfaction although all your arguments have been drawn upon the proposition that high job demands leads to burnout consequences. There is no rationale and explanation to this part as well.

Research model figure 1 doesn’t reflect your study’s hypothetical paths. Why this model is showing the job engagement as mediator between job demands and job satisfaction while burnout between job resources and satisfaction. Though your hypothesis focuses on the loss and gain paths only. Moreover, the interaction path is also missing. Redraw your model based on the proposed hypothetical links

Methodology/Analysis:

No convergent/discriminant validity and reliability assessment or CFA in an alternative case, are being carried out to assess its fit indices or measurement model adequacy.

Several inter-construct correlation values are above 0.60, posing serious concern over the presence of multi-collinearity issues among the variables. Need to re-check them via VIF. There might be some discriminant validity issues as well. Redo this part

H1 hypothesis results proposing the direct path to job satisfaction is missing…………?

Mediation testing is ambiguous and confusing too. In model 1 and 2 of table 4 beta coefficients have been reported while the rest of the models t-values are reported, why? Most importantly you proposed job demands and resources as a composite construct comprising of sub-dimensions throughout your manuscript, however, you have directly tested the effect of sub-dimensions on the outcome variables. It is again showing a serious limitation over the understanding about the objectivity and purpose of your study. Be focused on what actually you are intended to examine.

I suggest to further simplify it by using Hayes process macro to test your mediation result

Interaction effect results and table is missing too…………?

Testing job satisfaction profiles (optimization level) is beyond the scope of your study, you haven’t argued and discuss it in your introduction and hypothesis development section. Why all of sudden it comes to analysis? I would suggest, either you should focus on the mediation and interaction hypothesis or may only choose to build your manuscript around job satisfaction profiling (degree of low-high proposition) assessment.

Discussion:

Discussion section is reasonable compared to rest of the manuscript. I would suggest to build your hypothesis development section in view of your discussion section. Importantly, your discussion is all about the varying degree of influence of sub-dimensions to outcome variables therefore, your hypothesis development should consider all as separate variables then.

We look forward to receiving your revised manuscript.

Kind regards,

Muhammad Fareed, Ph.D

Academic Editor

PLOS ONE

Journal Requirements:

Additional Editor Comments

Dear Author,

Please make the amendments as per the followings:

Reviewer 1

This paper examines the Job Demands,Job Resources, and Postdoctoral Job Satisfaction: An Empirical Study Based on the Data from Nature’s Global Postdoc Survey. The paper is in general not satisfactorily written and lacks novelty and justification for the need of study. I would like to make a few suggestions to improve the contributions of this research.

Abstract

The abstract should reflect the snapshot of the paper. It should be more persuasive. The Abstract lacks originality. Revisit the abstract specifically focusing on the following steps; (Brief glimpse into the topic, Purpose, Methodology, and Practical Implications).

Introduction

The Mechanism of introduction is not appropriate. It must include the territory of the study, niche, and detail about the current research, leading toward the need for this research. It lacks justification for; the need for the study, the selection of the variables, and the relevance of the population/context of the study. Furthermore, the introduction lacks to justify the contribution of this study to Academia and/or Industry, why is it worth studying? What`s new in it and why does it matter? In short, the entire introduction Part of the chapter needs serious attention.

Literature Review

Under the heading "1. Two-path hypothesis of the influence of job characteristics on post-doctoral job satisfaction” It is mentioned that earlier, studies have been conducted on the same topic in the same context, then the question arises, “What the current study is going to add in it??

A few references are NOT in APA style, as per the 7th edition. See, For example, on page 10, the last line of the second paragraph.

What do you mean by this term “In the university field? at page 10

Hypothesis H4 is contradictory to Hypothesis H2. H2 states that job demands have a negative indirect effect on postdoctoral job satisfaction……. While H4 states that “Postdoctoral job satisfaction is highest with the combination of “high demands-high resources”. Means positively associated…?? Kindly elaborate on how??

This section contains many grammatical mistakes and typo errors. Kindly proofread.

The scholar has failed to develop a link between IVs and DV. A number of claims require referencing. Hence, the Literature requires significant rework.

Research Methodology

Data collection methods, inclusion-exclusion criteria, Justification for sample size, etc. are not properly discussed.

It is mentioned that “The right questions from the respective portions of the questionnaire were chosen to describe the variables being looked at". HOW??? Is any specific method applied?? Discuss its justification as well.

No explanation of instrumentation is mentioned. E.g. job satisfaction questionnaire is adopted/adapted from?? Or was either developed??

Statistical Methods and Results

Why SPSS is used?? And if SPSS, then why the quite older version is used since we have many revisions after SPSS 22.0? Instead, it is recommended to use SEM and SMART-PLS.

Why two-factor ANOVA is used for mediation??? Can you provide any logical justification for it?

“Where the regression coefficients are explored in a two-path test with maximum likelihood estimation; and in the test of mediating and moderating effects using the self-help method (Bootstrap Method), What is this? Where is the moderator in the entire study?? How mediation is checked through the self-help method? Etc.

If SPSS, is used, then why not “Baron and Kenny” OR “Andrew Hayes approach” applied for mediation?

“The interaction hypothesis of job demands and job resources on postdoctoral job satisfaction” What is the need for all this?? Elaborate on the justification of its relevance.

“Table 7 Self-help method test for the mediated role of the moderated” What does this table indicate???

What about the normality of data? Any test applied?? discuss

The entire section does not have a single reference. The author is supposed to support its results from existing literature.

In short, this section requires some serious attention.

Discussion and Conclusion

Discussion should relate or contradict the study finding in light of existing research furthermore why your study found such results; there should be a discussion on results rather than just confirming the results.

The conclusion of the study should be drafted in light of the objectives of the study. The author should highlight how each of the objectives identified in the research was attained.

Final Evaluation Statement:

The paper contains serious issues regarding, the novelty of the paper, justification of the study, methodology applied, and analysis section, besides, formatting issues like language, sequence, spelling, grammar, and coherence).

Reviewer 2

Introduction:

Introduction section focuses primarily on presenting facts and figures concerning the previous studies. It lacks the originality, significance, and contribution of your study. Why your conceptual framework is important to be studied, and how well it articulates the overall purpose of your study? The authors should highlight the role of job demands, resources in affecting the job satisfaction and why it is critical to examine motivational and hindrance mechanism to have deeper insights about the topic. Meanwhile, synchronize your framework with the overarching theory (i.e. JD-R) comprising the conceptual base of your study.

Literature Review:

Literature review section lacks clarity, coherence and relevance. For example, line 7 -12 of first para is supposed to outline two dimensions, however there are so many multi-factors have been highlighted without any due clarity of what your study aims to examine? It seems lack of understanding and objectivity. Likewise, putting citation (i.e. Weiss, Dawis, & England, 1967) in the end doesn’t sound good as it shows your study aims pointing to someone else work.

In second para, you stated that “After controlling for demographic variables, the research on the influencing factors of postdoctoral job satisfaction is primarily focused on a single dimension…………” Which research you are referring about? Please cite it with substantive evidence to support your claim.

In same para, you stated that “Second, the study lacks a better theoretical framework for evaluating and reflecting on postdoctoral job satisfaction from a systematic standpoint…………” which study lacks better theoretical framework? You need to revisit your literature review section it lacks substantive arguments and also have some technical writing issues.

Research Hypothesis:

In hypothesis section, you stated “The first hypothesis is the “dual path” hypothesis, which states that “there are two ways in which employment influences employees: the loss path and the gain path” however, it doesn’t make a sense to test the direct effect of job demands and resources on job satisfaction and call it the loss and gain path. Loss and gain paths were originally formulated to test outcome variables through job engagement and burnout, however testing direct paths to job satisfaction overrule such assumption. Revisit your statement and arguments.

In similar way, your second hypothesis uncovers …….“the buffering hypothesis, which states that job resources can mitigate the negative effects of job demands by mitigating the attrition of employees with high job demands”…….. however, in hypothesis section 2, you hypothesized the mediating role of burnout and engagement. Again it is misleading and vague. Revisit

Hypothesis section 3 aims to examine interaction effect of job resources and demands on job satisfaction. How comes the interaction effect would be carried out on the job satisfaction? Moreover, hypothesis H3a hypothesizes…… “job demands can enhance the positive impact of job resources on postdoctoral work engagement” …… How job demands can enhance this relationship? You haven’t provided any valid justification to this claim. Explain it

Hypothesis 4 is misleading too. How high demand-high resource combination produces the optimum satisfaction although all your arguments have been drawn upon the proposition that high job demands leads to burnout consequences. There is no rationale and explanation to this part as well.

Research model figure 1 doesn’t reflect your study’s hypothetical paths. Why this model is showing the job engagement as mediator between job demands and job satisfaction while burnout between job resources and satisfaction. Though your hypothesis focuses on the loss and gain paths only. Moreover, the interaction path is also missing. Redraw your model based on the proposed hypothetical links

Methodology/Analysis:

No convergent/discriminant validity and reliability assessment or CFA in an alternative case, are being carried out to assess its fit indices or measurement model adequacy.

Several inter-construct correlation values are above 0.60, posing serious concern over the presence of multi-collinearity issues among the variables. Need to re-check them via VIF. There might be some discriminant validity issues as well. Redo this part

H1 hypothesis results proposing the direct path to job satisfaction is missing…………?

Mediation testing is ambiguous and confusing too. In model 1 and 2 of table 4 beta coefficients have been reported while the rest of the models t-values are reported, why? Most importantly you proposed job demands and resources as a composite construct comprising of sub-dimensions throughout your manuscript, however, you have directly tested the effect of sub-dimensions on the outcome variables. It is again showing a serious limitation over the understanding about the objectivity and purpose of your study. Be focused on what actually you are intended to examine.

I suggest to further simplify it by using Hayes process macro to test your mediation result

Interaction effect results and table is missing too…………?

Testing job satisfaction profiles (optimization level) is beyond the scope of your study, you haven’t argued and discuss it in your introduction and hypothesis development section. Why all of sudden it comes to analysis? I would suggest, either you should focus on the mediation and interaction hypothesis or may only choose to build your manuscript around job satisfaction profiling (degree of low-high proposition) assessment.

Discussion:

Discussion section is reasonable compared to rest of the manuscript. I would suggest to build your hypothesis development section in view of your discussion section. Importantly, your discussion is all about the varying degree of influence of sub-dimensions to outcome variables therefore, your hypothesis development should consider all as separate variables then.

Reviewers' comments:

Reviewer's Responses to Questions

**Comments to the Author**

1. Is the manuscript technically sound, and do the data support the conclusions?

Reviewer #1: Partly

Reviewer #2: Partly

2. Has the statistical analysis been performed appropriately and rigorously? 

Reviewer #1: No

Reviewer #2: No

3. Have the authors made all data underlying the findings in their manuscript fully available?

Reviewer #1: Yes

Reviewer #2: No

4. Is the manuscript presented in an intelligible fashion and written in standard English?

Reviewer #1: Yes

Reviewer #2: Yes

5. Review Comments to the Author

Reviewer #1: This paper examines the Job Demands,Job Resources, and Postdoctoral Job Satisfaction: An Empirical Study Based on the Data from Nature’s Global Postdoc Survey. The paper is in general not satisfactorily written and lacks novelty and justification for the need of study. I would like to make a few suggestions to improve the contributions of this research.

Abstract

The abstract should reflect the snapshot of the paper. It should be more persuasive. The Abstract lacks originality. Revisit the abstract specifically focusing on the following steps; (Brief glimpse into the topic, Purpose, Methodology, and Practical Implications).

Introduction

The Mechanism of introduction is not appropriate. It must include the territory of the study, niche, and detail about the current research, leading toward the need for this research. It lacks justification for; the need for the study, the selection of the variables, and the relevance of the population/context of the study. Furthermore, the introduction lacks to justify the contribution of this study to Academia and/or Industry, why is it worth studying? What`s new in it and why does it matter? In short, the entire introduction Part of the chapter needs serious attention.

Literature Review

Under the heading "1. Two-path hypothesis of the influence of job characteristics on post-doctoral job satisfaction” It is mentioned that earlier, studies have been conducted on the same topic in the same context, then the question arises, “What the current study is going to add in it??

A few references are NOT in APA style, as per the 7th edition. See, For example, on page 10, the last line of the second paragraph.

What do you mean by this term “In the university field? at page 10

Hypothesis H4 is contradictory to Hypothesis H2. H2 states that job demands have a negative indirect effect on postdoctoral job satisfaction……. While H4 states that “Postdoctoral job satisfaction is highest with the combination of “high demands-high resources”. Means positively associated…?? Kindly elaborate on how??

This section contains many grammatical mistakes and typo errors. Kindly proofread.

The scholar has failed to develop a link between IVs and DV. A number of claims require referencing. Hence, the Literature requires significant rework.

Research Methodology

Data collection methods, inclusion-exclusion criteria, Justification for sample size, etc. are not properly discussed.

It is mentioned that “The right questions from the respective portions of the questionnaire were chosen to describe the variables being looked at". HOW??? Is any specific method applied?? Discuss its justification as well.

No explanation of instrumentation is mentioned. E.g. job satisfaction questionnaire is adopted/adapted from?? Or was either developed??

Statistical Methods and Results

Why SPSS is used?? And if SPSS, then why the quite older version is used since we have many revisions after SPSS 22.0? Instead, it is recommended to use SEM and SMART-PLS.

Why two-factor ANOVA is used for mediation??? Can you provide any logical justification for it?

“Where the regression coefficients are explored in a two-path test with maximum likelihood estimation; and in the test of mediating and moderating effects using the self-help method (Bootstrap Method), What is this? Where is the moderator in the entire study?? How mediation is checked through the self-help method? Etc.

If SPSS, is used, then why not “Baron and Kenny” OR “Andrew Hayes approach” applied for mediation?

“The interaction hypothesis of job demands and job resources on postdoctoral job satisfaction” What is the need for all this?? Elaborate on the justification of its relevance.

“Table 7 Self-help method test for the mediated role of the moderated” What does this table indicate???

What about the normality of data? Any test applied?? discuss

The entire section does not have a single reference. The author is supposed to support its results from existing literature.

In short, this section requires some serious attention.

Discussion and Conclusion

Discussion should relate or contradict the study finding in light of existing research furthermore why your study found such results; there should be a discussion on results rather than just confirming the results.

The conclusion of the study should be drafted in light of the objectives of the study. The author should highlight how each of the objectives identified in the research was attained.

Final Evaluation Statement:

The paper contains serious issues regarding, the novelty of the paper, justification of the study, methodology applied, and analysis section, besides, formatting issues like language, sequence, spelling, grammar, and coherence).

Reviewer #2: Introduction:

Introduction section focuses primarily on presenting facts and figures concerning the previous studies. It lacks the originality, significance, and contribution of your study. Why your conceptual framework is important to be studied, and how well it articulates the overall purpose of your study? The authors should highlight the role of job demands, resources in affecting the job satisfaction and why it is critical to examine motivational and hindrance mechanism to have deeper insights about the topic. Meanwhile, synchronize your framework with the overarching theory (i.e. JD-R) comprising the conceptual base of your study.

Literature Review:

Literature review section lacks clarity, coherence and relevance. For example, line 7 -12 of first para is supposed to outline two dimensions, however there are so many multi-factors have been highlighted without any due clarity of what your study aims to examine? It seems lack of understanding and objectivity. Likewise, putting citation (i.e. Weiss, Dawis, & England, 1967) in the end doesn’t sound good as it shows your study aims pointing to someone else work.

In second para, you stated that “After controlling for demographic variables, the research on the influencing factors of postdoctoral job satisfaction is primarily focused on a single dimension…………” Which research you are referring about? Please cite it with substantive evidence to support your claim.

In same para, you stated that “Second, the study lacks a better theoretical framework for evaluating and reflecting on postdoctoral job satisfaction from a systematic standpoint…………” which study lacks better theoretical framework? You need to revisit your literature review section it lacks substantive arguments and also have some technical writing issues.

Research Hypothesis:

In hypothesis section, you stated “The first hypothesis is the “dual path” hypothesis, which states that “there are two ways in which employment influences employees: the loss path and the gain path” however, it doesn’t make a sense to test the direct effect of job demands and resources on job satisfaction and call it the loss and gain path. Loss and gain paths were originally formulated to test outcome variables through job engagement and burnout, however testing direct paths to job satisfaction overrule such assumption. Revisit your statement and arguments.

In similar way, your second hypothesis uncovers …….“the buffering hypothesis, which states that job resources can mitigate the negative effects of job demands by mitigating the attrition of employees with high job demands”…….. however, in hypothesis section 2, you hypothesized the mediating role of burnout and engagement. Again it is misleading and vague. Revisit

Hypothesis section 3 aims to examine interaction effect of job resources and demands on job satisfaction. How comes the interaction effect would be carried out on the job satisfaction? Moreover, hypothesis H3a hypothesizes…… “job demands can enhance the positive impact of job resources on postdoctoral work engagement” …… How job demands can enhance this relationship? You haven’t provided any valid justification to this claim. Explain it

Hypothesis 4 is misleading too. How high demand-high resource combination produces the optimum satisfaction although all your arguments have been drawn upon the proposition that high job demands leads to burnout consequences. There is no rationale and explanation to this part as well.

Research model figure 1 doesn’t reflect your study’s hypothetical paths. Why this model is showing the job engagement as mediator between job demands and job satisfaction while burnout between job resources and satisfaction. Though your hypothesis focuses on the loss and gain paths only. Moreover, the interaction path is also missing. Redraw your model based on the proposed hypothetical links

Methodology/Analysis:

No convergent/discriminant validity and reliability assessment or CFA in an alternative case, are being carried out to assess its fit indices or measurement model adequacy.

Several inter-construct correlation values are above 0.60, posing serious concern over the presence of multi-collinearity issues among the variables. Need to re-check them via VIF. There might be some discriminant validity issues as well. Redo this part

H1 hypothesis results proposing the direct path to job satisfaction is missing…………?

Mediation testing is ambiguous and confusing too. In model 1 and 2 of table 4 beta coefficients have been reported while the rest of the models t-values are reported, why? Most importantly you proposed job demands and resources as a composite construct comprising of sub-dimensions throughout your manuscript, however, you have directly tested the effect of sub-dimensions on the outcome variables. It is again showing a serious limitation over the understanding about the objectivity and purpose of your study. Be focused on what actually you are intended to examine.

I suggest to further simplify it by using Hayes process macro to test your mediation result

Interaction effect results and table is missing too…………?

Testing job satisfaction profiles (optimization level) is beyond the scope of your study, you haven’t argued and discuss it in your introduction and hypothesis development section. Why all of sudden it comes to analysis? I would suggest, either you should focus on the mediation and interaction hypothesis or may only choose to build your manuscript around job satisfaction profiling (degree of low-high proposition) assessment.

Discussion:

Discussion section is reasonable compared to rest of the manuscript. I would suggest to build your hypothesis development section in view of your discussion section. Importantly, your discussion is all about the varying degree of influence of sub-dimensions to outcome variables therefore, your hypothesis development should consider all as separate variables then.

6. PLOS authors have the option to publish the peer review history of their article (what does this mean?). If published, this will include your full peer review and any attached files.

Reviewer #1: **Yes: **Muhammad Shahid Shams

Reviewer #2: **Yes: **Dr. Farhan Mehboob

---

## [Author Response · Author response to Decision Letter 0]

24 Aug 2023

We are very thankful to the editor for giving us chance to revise our paper. We also thank to all reviewers for their constructive feedback and comments to improve the quality of this paper. The authors have modified the paper according to the reviewers’ comments and provided line by line response to their comments and reference to changes in the revised paper. If the reviewer has new comments on the revised manuscript, we will revise it again.

Replies to the comments of Reviewer # 1

Abstract

The abstract should reflect the snapshot of the paper. It should be more persuasive. The Abstract lacks originality. Revisit the abstract specifically focusing on the following steps; (Brief glimpse into the topic, Purpose, Methodology, and Practical Implications).

Reply: Thank you very much for the valuable suggestion. We have reorganized the content of the abstract based on your comments to include: Research background, research purpose, research methodology, results, and contribution of the study. For details, please refer to L#11-25of page 1.

Introduction

The Mechanism of introduction is not appropriate. It must include the territory of the study, niche, and detail about the current research, leading toward the need for this research. It lacks justification for; the need for the study, the selection of the variables, and the relevance of the population/context of the study. Furthermore, the introduction lacks to justify the contribution of this study to Academia and/or Industry, why is it worth studying? What`s new in it and why does it matter? In short, the entire introduction Part of the chapter needs serious attention. 

Reply: Thank you very much for the valuable suggestion. We have rewritten the introduction section and added the research significance at the end of this part. The ideas in the introduction section are: the importance of the postdoctoral system, the poor status of postdoctoral job satisfaction, the contributions and shortcomings of existing research, the need for the study, the methodology and purpose of the study. For details, please refer to L#28-38 of page 1, L#1-29of page 2.

Previous research results help us to understand the current status of postdoctoral satisfaction and its influencing factors, but the relationship between the influencing factors has not been reasonably explained and clarified, and a better theoretical framework is lacking. Therefore, the innovation of this study is to depict the influencing factors and and the intrinsic mechanism of postdoctoral job satisfaction in a more in-depth, detailed, comprehensive and holistic way by using the JD-R model from the postdoctoral work field and the characteristic factors in it. If there is anything we need to explain clearly, we would like to ask the reviewers to point it out, and we are willing to accept and make changes.

Literature Review

Under the heading "1. Two-path hypothesis of the influence of job characteristics on post-doctoral job satisfaction” It is mentioned that earlier, studies have been conducted on the same topic in the same context, then the question arises, “What the current study is going to add in it?? 

Reply: Thank you very much for the valuable suggestion. Existing studies have measured and confirmed the predictive role of certain types of job resources or job demands on postdoctoral job satisfaction from a micro perspective, respectively, but have not yet analyzed the role of job resources and job requirements on postdoctoral job satisfaction from a holistic perspective.

For details, please refer to L#1-10 of page 6.

A few references are NOT in APA style, as per the 7th edition. See, For example, on page 10, the last line of the second paragraph.

Reply: We apologize for the reference formatting errors in the manuscript and thank you for the correction, we have checked and corrected all reference formats.

What do you mean by this term “In the university field? at page 10

Reply: We apologize for not being clear in the manuscript. We would like to express that in the area of postdoctoral job satisfaction research, we have corrected the statement in the manuscript.

Hypothesis H4 is contradictory to Hypothesis H2. H2 states that job demands have a negative indirect effect on postdoctoral job satisfaction……. While H4 states that “Postdoctoral job satisfaction is highest with the combination of “high demands-high resources”. Means positively associated…?? Kindly elaborate on how??

Reply: Thank you very much for the valuable suggestion. Initially, we hope to combine the Buffering Hypothesis with the Coping Hypothesis to further infer that the optimal ratio of job demands to job resources. However, after careful consideration, we think it does lack sufficient theoretical support and is not sufficiently connected to the subject matter,so we have have deleted Hypothesis H4 according to the reviewer' comments, we also apologize for the uncomprehensive thinking before.

This section contains many grammatical mistakes and typo errors. Kindly proofread. 

Reply: We apologize for the grammatical mistakes and typo errors in the manuscript and thank you for the correction, we have rechecked grammatical errors in the manuscript.

The scholar has failed to develop a link between IVs and DV. A number of claims require referencing. Hence, the Literature requires significant rework.

Reply: We rewrote the literature review section around the three dimensions: connotation of job satisfaction, the current status of postdoctoral job satisfaction, and the influencing factors of postdoctoral job satisfaction. And we reorganized the research hypothesis section and added relevant references. If the reviewers believe there is a more appropriate way for this part, we will adopt the reviewer’ suggestions. 

Research Methodology

Data collection methods, inclusion-exclusion criteria, Justification for sample size, etc. are not properly discussed.

Reply: Thank you for your comments. The data was collected by Nature, after excluding non-postdoctoral staff, part-time postdoctoral staff, and missing values and invalid samples, 5,861 samples were retained in the final screening(L#16-17 of page 8). The Cronbach’s alpha coefficients for each variable reached the acceptable level of 0.7 suggested by the study, indicating good reliability(L#43-44 of page 9). We further examined the discriminant validity of the four dimensions using average extracted variance (AVE). The AVE values for each dimension ranged from 0.60 to 0.67, which were above the critical value of 0.5, indicating good discriminant validity(L#3-8 of page 13). If there are any other questions about the data source, please point out and we will continue to supplement and modify it.

It is mentioned that “The right questions from the respective portions of the questionnaire were chosen to describe the variables being looked at". HOW??? Is any specific method applied?? Discuss its justification as well.

Reply: Thank you for your comments. We discuss how to select the appropriate questions from the questionnaire in the Measurement indicators section(page 8-page 9). Since the questionnaire was not developed by us, we selected relevant questions from the questionnaire that could represent the job resources and job requirements, taking into account the JD-R model, other relevant theories and previous studies.

No explanation of instrumentation is mentioned. E.g. job satisfaction questionnaire is adopted/adapted from?? Or was either developed??

Reply: The postdoctoral job satisfaction metric (question 43: How satisfied are you with your current postdoctoral work?) was the dependent variable(L#24-27 of page 8). Other measurement indicators are based on the JD-R model, combined with the practical situation of postdoctoral work and existing literature classification(page 8-page 9).

Statistical Methods and Results

Why SPSS is used?? And if SPSS, then why the quite older version is used since we have many revisions after SPSS 22.0? Instead, it is recommended to use SEM and SMART-PLS.

Reply: Thank you very much for the valuable suggestion. Initially,we used SPSS 22.0 , aiming to use the Process plug-in developed by Hayes in SPSS to test for mediation and moderation effects. Now we have reprocessed the data using SPSS27.0 and added AMOS26.0 for confirmatory factor analysis(L#2-15 of page 11).

Why two-factor ANOVA is used for mediation??? Can you provide any logical justification for it?

Reply: We apologize for the mistake in the manuscript. We rewrote the research methods section，the regression analyses were conducted to examine the hypothesis 2 and hypothesis 3(L#2-15 of page 11).

“Where the regression coefficients are explored in a two-path test with maximum likelihood estimation; and in the test of mediating and moderating effects using the self-help method (Bootstrap Method), What is this? Where is the moderator in the entire study?? How mediation is checked through the self-help method? Etc.

Reply: Thank you very much for the valuable suggestion. We've reorganized this section. The SPSS moderation mediation testing algorithm follows the Process program developed by Andrew F. Hayes, and we use the moderation mediation testing model in SPSS to test hypothesis 3. if the difference in the conditional indirect effect size of the mediated role in both cases does not contain 0 at the 95% confidence interval when the moderating variable W is considered to be high (1 standard deviation above the mean) or low (1 standard deviation below the mean), respectively, the mediated role of the moderated role can be assumed to be supported(Hayes, 2017).

In Hypothesis H3, job resources play a moderating role in the pathway in which job demands affect job satisfaction through job burnout, and similarly, job demands play a moderating role in the pathway in which job resources affect job satisfaction through job engagement, so job resources and job demands are the moderators in both pathways(L#6-31 of page 7).

If SPSS, is used, then why not “Baron and Kenny” OR “Andrew Hayes approach” applied for mediation?

Reply: Thank you very much for the valuable suggestion. We retested the mediating effect of hypothesis H2 using Andrew Hayes approach, and the results are shown in Table 6(L# 25 of page 14). 

“The interaction hypothesis of job demands and job resources on postdoctoral job satisfaction” What is the need for all this?? Elaborate on the justification of its relevance. 

Reply: We have have deleted Hypothesis H4 and the relevant hypothesis testing content according to the reviewer' comments, we also apologize for the uncomprehensive thinking before.

“Table 7 Self-help method test for the mediated role of the moderated” What does this table indicate???

Reply: Table 7 shows the moderated mediation model test results. We have refined the research process of this section (L#15-18 of page 15).We use the moderation mediation testing model in SPSS to test hypothesis 3. In order to reduce multicollinearity, the independent and moderating variables were first centered and grouped according to the moderating variables according to (M±1 std)，if the difference in the conditional indirect effect size of the mediated role in both cases does not contain 0 at the 95% confidence interval when the moderating variable is considered to be high (1 standard deviation above the mean) or low (1 standard deviation below the mean), respectively, the mediated role of the moderated role can be assumed to be supported(Hayes, 2017). According to table 7 , hypotheses H3a and H3b are supported on the life demands dimension and the organizational and humanistic resource dimensions. We are open to comments if the reviewer think the tables need to be revised more.

What about the normality of data? Any test applied?? discuss

Reply: Thank you very much for the valuable suggestion. We have added VIF(L#2-5 of page 12), AVE(L#3-8 of page 13), and CFA test (L#8-14 of page 12)in the manuscript. If there is anything we need to explain clearly, we would like to ask the reviewers to point it out, and we are willing to accept and make changes.

The entire section does not have a single reference. The author is supposed to support its results from existing literature.

Reply: We agree with the comment and have added relevant references in this section. (L#5 of page 12, L#10-11 of page 12, L#3-8 of page 13, L#7-11 of page 14) 

In short, this section requires some serious attention.

Reply: Thank you very much for the valuable suggestion.We have adjusted and modified this part, supplemented CFA, VIF and other related tests, and added relevant references, if there are still inappropriate places please continue to criticize, we will continue to modify.

Discussion and Conclusion 

Discussion should relate or contradict the study finding in light of existing research furthermore why your study found such results; there should be a discussion on results rather than just confirming the results. 

Reply: Thank you very much for the valuable suggestion.We have revised the discussion section significantly, comparing it more with studies in a global context. If there is anything we need to explain clearly, we would like to ask the reviewers to point it out, and we are willing to accept and make changes.

The conclusion of the study should be drafted in light of the objectives of the study. The author should highlight how each of the objectives identified in the research was attained. 

Reply: Thank you very much for the valuable suggestion. We have reorganized the conclusion section, highlighted how research objectives are achieved and the research contributions (L#10-29 of page 19) .

. 

Replies to the comments of Reviewer # 2

We are very grateful for the comments given by the reviewer, which were very professional and helpful. We have accepted all the reviewer' comments and revised the manuscript. If the reviewer has new comments on the revised manuscript, we will revise it again.

Introduction:Introduction section focuses primarily on presenting facts and figures concerning the previous studies. It lacks the originality, significance, and contribution of your study. Why your conceptual framework is important to be studied, and how well it articulates the overall purpose of your study? The authors should highlight the role of job demands, resources in affecting the job satisfaction and why it is critical to examine motivational and hindrance mechanism to have deeper insights about the topic. Meanwhile, synchronize your framework with the overarching theory (i.e. JD-R) comprising the conceptual base of your study.

Reply: Thank you very much for the valuable suggestion. We have rewritten the introduction section and added the research significance and contribution at the end of this part. The ideas in the introduction section are: the importance of the postdoctoral system, the poor status of postdoctoral job satisfaction, the contributions and shortcomings of existing research, the need for the study, the methodology and purpose of the study. For details, please refer to L#28-38 of page 1, L#1-29of page 2.

Previous research results help us to understand the current status of postdoctoral satisfaction and its influencing factors, but the relationship between the influencing factors has not been reasonably explained and clarified, and a better theoretical framework is lacking. Therefore, the innovation of this study is to depict the influencing factors and and the intrinsic mechanism of postdoctoral job satisfaction in a more in-depth, detailed, comprehensive and holistic way by using the JD-R model from the postdoctoral work field and the characteristic factors in it. If the reviewers believe there is a more appropriate way for this part, we will adopt the reviewer’ suggestions.

Literature Review: Literature review section lacks clarity, coherence and relevance. For example, line 7 -12 of first para is supposed to outline two dimensions, however there are so many multi-factors have been highlighted without any due clarity of what your study aims to examine? It seems lack of understanding and objectivity. Likewise, putting citation (i.e. Weiss, Dawis, & England, 1967) in the end doesn’t sound good as it shows your study aims pointing to someone else work. In second para, you stated that “After controlling for demographic variables, the research on the influencing factors of postdoctoral job satisfaction is primarily focused on a single dimension…………” Which research you are referring about? Please cite it with substantive evidence to support your claim. In same para, you stated that “Second, the study lacks a better theoretical framework for evaluating and reflecting on postdoctoral job satisfaction from a systematic standpoint…………” which study lacks better theoretical framework? You need to revisit your literature review section it lacks substantive arguments and also have some technical writing issues.

Reply: Thank you very much for the valuable suggestion.We rewrote the literature review section around the three dimensions: connotation of job satisfaction, the current status of postdoctoral job satisfaction, and the influencing factors of postdoctoral job satisfaction. Existing studies have confirmed that elements of job characteristics are closely related to postdoctoral job satisfaction, indicating that the JD-R model has a strong explanatory capacity for postdoctoral job satisfaction. However, the positive and negative effects of job characteristic elements on job satisfaction have not been organically combined to form a balanced and comprehensive analytic framework with interaction. Based on this, the JD-R model is selected as the analytical perspective. If the reviewers believe there is a more appropriate way for this part, we will adopt the reviewer' suggestions.

For details, please refer to L#33-40 of page 2, page 3-4.

Research Hypothesis:In hypothesis section, you stated “The first hypothesis is the “dual path” hypothesis, which states that “there are two ways in which employment influences employees: the loss path and the gain path” however, it doesn’t make a sense to test the direct effect of job demands and resources on job satisfaction and call it the loss and gain path. Loss and gain paths were originally formulated to test outcome variables through job engagement and burnout, however testing direct paths to job satisfaction overrule such assumption. Revisit your statement and arguments.In similar way, your second hypothesis uncovers …….“the buffering hypothesis, which states that job resources can mitigate the negative effects of job demands by mitigating the attrition of employees with high job demands”…….. however, in hypothesis section 2, you hypothesized the mediating role of burnout and engagement. Again it is misleading and vague. Revisit

Reply: We apologize for not being clear in the manuscript. We have revised and added relevant content about job resources and job demands directly affecting job satisfaction.The logic is that an increase in job resources such as performance feedback, salary and compensation, and autonomy in decision-making can directly satisfy the needs of employees’ competence and autonomy, provide job security for individuals, promote their positive perception of job satisfaction. Increasing job demands consume employees’ resources, leading to a loss of resources, which may directly lead to health problems or negative perceptions of job satisfaction, forming the “attrition path” of job satisfaction. If there is anything we need to explain clearly, we would like to ask the reviewers to point it out, and we are willing to accept and make changes. 

For details, please refer to L#25-42 of page 5.

Hypothesis section 3 aims to examine interaction effect of job resources and demands on job satisfaction. How comes the interaction effect would be carried out on the job satisfaction? Moreover, hypothesis H3a hypothesizes…… “job demands can enhance the positive impact of job resources on postdoctoral work engagement” …… How job demands can enhance this relationship? You haven’t provided any valid justification to this claim. Explain it

Reply: We apologize for not being clear in the manuscript and added relevant content. On the one hand, job demands are more likely to be met in a more resourceful environment, which reduces the likelihood of job demands causing job stress and buffers the negative effects of high job demands on individuals, also known as the“buffering hypothesis”; on the other hand, in a more demanding environment, individuals are less likely to be inert, and are able to engage more fully in their work, mobilizing the large amount of work resources they already have to acquit the demands of the environment. This is also known as the “coping hypothesis”.We have added some substantial investigations in the manuscript to provide support for this hypothesis. If there is anything we need to explain clearly, we would like to ask the reviewers to point it out, and we are willing to accept and make changes.

For details, please refer to L#6-26 of page 7.

Hypothesis 4 is misleading too. How high demand-high resource combination produces the optimum satisfaction although all your arguments have been drawn upon the proposition that high job demands leads to burnout consequences. There is no rationale and explanation to this part as well.

Reply: Thank you very much for the valuable suggestion. Initially, we hope to combine the Buffering Hypothesis with the Coping Hypothesis to further infer that the optimal ratio of job demands to job resources. However, after careful consideration, and based on your comments ,we think we should focus our research on the relationship between job factors and job satisfaction, so we have deleted Hypothesis H4 and related Analysis，we also apologize for the uncomprehensive thinking before.

Research model figure 1 doesn’t reflect your study’s hypothetical paths. Why this model is showing the job engagement as mediator between job demands and job satisfaction while burnout between job resources and satisfaction. Though your hypothesis focuses on the loss and gain paths only. Moreover, the interaction path is also missing. Redraw your model based on the proposed hypothetical links

Reply: Thank you very much for the valuable suggestion. We have redraw our model 1. For details, please refer to L#1 of page 8.

Methodology/Analysis:No convergent/discriminant validity and reliability assessment or CFA in an alternative case, are being carried out to assess its fit indices or measurement model adequacy.

Reply: Thank you very much for the valuable suggestion. We supplemented our CFA with AMOS 26.0. The results are shown in table 2. The 4-factor model (X2=467.77, df=179, X2/df=2.613, CFI=0.963, RMSEA=0.058, TLI=0.943) fits better than all the other models and has a good matching index, which implies that there is very good discriminative validity among the four variables in this study. For details, please refer to L#8-14 of page 12.

Several inter-construct correlation values are above 0.60, posing serious concern over the presence of multi-collinearity issues among the variables. Need to re-check them via VIF. There might be some discriminant validity issues as well. Redo this part

Reply: Thank you very much for the valuable suggestion. We tested the data for the existence of multicollinearity using VIF, and the results are shown in Table 3. The maximum VIF between the variables is 2.180, which is less than 10, and the minimum tolerance is 0.459, which is greater than 0.1, so it can be seen that there is no serious problem of multicollinearity between the variables. For details, please refer to L#2-5 of page 12.

H1 hypothesis results proposing the direct path to job satisfaction is missing…………?

Reply: Thank you very much for the valuable suggestion. According to the means and correlation coefficients of the variables depicted in table 5, showing that job demands are all significantly negatively associated with job satisfaction, whereas job resources are all significantly positively associated with job satisfaction. For details, please refer to page 13.

Mediation testing is ambiguous and confusing too. In model 1 and 2 of table 4 beta coefficients have been reported while the rest of the models t-values are reported, why? Most importantly you proposed job demands and resources as a composite construct comprising of sub-dimensions throughout your manuscript, however, you have directly tested the effect of sub-dimensions on the outcome variables. It is again showing a serious limitation over the understanding about the objectivity and purpose of your study. Be focused on what actually you are intended to examine.

I suggest to further simplify it by using Hayes process macro to test your mediation result

Interaction effect results and table is missing too…………?

Reply: Thank you very much for the valuable suggestion. We retested the mediating effect of hypothesis H2 using Andrew Hayes approach, and the results are shown in Table 6.We are open to comments if the reviewer think the figures need to be revised more. For details, please refer to page 14.

Testing job satisfaction profiles (optimization level) is beyond the scope of your study, you haven’t argued and discuss it in your introduction and hypothesis development section. Why all of sudden it comes to analysis? I would suggest, either you should focus on the mediation and interaction hypothesis or may only choose to build your manuscript around job satisfaction profiling (degree of low-high proposition) assessment.

Reply: Thank you very much for the valuable suggestion. We think we should focus our research on the relationship between job factors and job satisfaction, so we have deleted Hypothesis H4 and related Analysis，we also apologize for the uncomprehensive thinking before.

Discussion:Discussion section is reasonable compared to rest of the manuscript. I would suggest to build your hypothesis development section in view of your discussion section. Importantly, your discussion is all about the varying degree of influence of sub-dimensions to outcome variables therefore, your hypothesis development should consider all as separate variables then.

Reply: Thank you very much for the valuable suggestion.We supplemented our research hypotheses with relevant content on sub-dimensions. For details, please refer to L#2-4 of page 5, L#6-10 of page 5 .

If there is anything we need to explain clearly, we would like to ask the reviewers to point it out, and we are willing to accept and make changes.

---

## [Decision Letter · Decision Letter 1]

11 Oct 2023

PONE-D-23-15985R1Job Demands，Job Resources and Postdoctoral Job Satisfaction：An Empirical Study Based on the Data from 2020 Nature’s Global Postdoc SurveyPLOS ONE

Dear Dr. XinXing Duan,

Thank you for submitting your manuscript to PLOS ONE. After careful consideration, we feel that it has merit but does not fully meet PLOS ONE’s publication criteria as it currently stands. Therefore, we invite you to submit a revised version of the manuscript that addresses the points raised during the review process.

Dear Author,Please make the ammendments as suggested by the both reviewers. Reviewer#1

The Authors have fully incorporated the comments suggested.

There is a minor issue in the interaction hypothesis development section. Like "H3b: Job resources can reduce the negative effect of job demands on postdoctoral burnout" is erroneously stated. It should be "Job resources can reduce the positive effect of job demands on postdoctoral burnout" as job demands are generally positively associated with the burnout outcomes.

Moreover, I would suggest to cite few lines from the discussion section of the paper referred below to further clarify the coping and gain spiral mechanism for your interaction hypothesis.

Mehboob, F., Othman, N., Fareed, M., & Raza, A. (2022). Change Appraisals and Job Crafting as Foundation to Inculcate Support for Change: A Dual Manifestation. Revista Brasileira de Gestão de Negócios, 24, 207-229.

Reviewer#2

I have completed the review of the article, and I can confirm that all the suggested changes have been diligently incorporated. However, there is one additional comment I would like to make here;

Please use the following article to justify the benchmark being met in confirmatory factor analysis for Cronbach's alpha, composite reliability, and AVE values:

Abboh, U. A., Majid, A. H., Fareed, M., & Abdussalaam, I. I. (2022). High-performance work practices lecturers’ performance connection: Does working condition matter? Management in Education, 0(0). https://doi.org/10.1177/08920206211051468

We look forward to receiving your revised manuscript.

Kind regards,

Muhammad Fareed, Ph.D

Academic Editor

PLOS ONE

Journal Requirements:

Additional Editor Comments:

Dear Author,

Please make the amendments as suggested by the reviewer.

Reviewers' comments:

Reviewer's Responses to Questions

**Comments to the Author**

1. If the authors have adequately addressed your comments raised in a previous round of review and you feel that this manuscript is now acceptable for publication, you may indicate that here to bypass the “Comments to the Author” section, enter your conflict of interest statement in the “Confidential to Editor” section, and submit your "Accept" recommendation.

Reviewer #1: All comments have been addressed

Reviewer #2: All comments have been addressed

2. Is the manuscript technically sound, and do the data support the conclusions?

Reviewer #1: Yes

Reviewer #2: Yes

3. Has the statistical analysis been performed appropriately and rigorously? 

Reviewer #1: Yes

Reviewer #2: Yes

4. Have the authors made all data underlying the findings in their manuscript fully available?

Reviewer #1: Yes

Reviewer #2: Yes

5. Is the manuscript presented in an intelligible fashion and written in standard English?

Reviewer #1: Yes

Reviewer #2: Yes

6. Review Comments to the Author

Reviewer #1: I have completed the review of the article, and I can confirm that all the suggested changes have been diligently incorporated. However, there is one additional comment I would like to make here;

Please use the following article to justify the benchmark being met in confirmatory factor analysis for Cronbach's alpha, composite reliability, and AVE values:

Abboh, U. A., Majid, A. H., Fareed, M., & Abdussalaam, I. I. (2022). High-performance work practices lecturers’ performance connection: Does working condition matter? Management in Education, 0(0). https://doi.org/10.1177/08920206211051468

Reviewer #2: The Authors have fully incorporated the comments suggested.

There is a minor issue in the interaction hypothesis development section. Like

"H3b: Job resources can reduce the negative effect of job demands on postdoctoral burnout" is erroneously stated. It should be "Job resources can reduce the positive effect of job demands on postdoctoral burnout" as job demands are generally positively associated with the burnout outcomes.

Moreover, I would suggest to cite few lines from the discussion section of the paper referred below to further clarify the coping and gain spiral mechanism for your interaction hypothesis.

Mehboob, F., Othman, N., Fareed, M., & Raza, A. (2022). Change Appraisals and Job Crafting as Foundation to Inculcate Support for Change: A Dual Manifestation. Revista Brasileira de Gestão de Negócios, 24, 207-229.

Rest is Ok. I'm wishing you all the best for your publication.

7. PLOS authors have the option to publish the peer review history of their article (what does this mean?). If published, this will include your full peer review and any attached files.

Reviewer #1: No

Reviewer #2: **Yes: **Dr. Farhan Mehboob

---

## [Author Response · Author response to Decision Letter 1]

17 Oct 2023

We are very thankful to the editor for giving us chance to revise our paper. We also thank to all reviewers for their constructive feedback and comments to improve the quality of this paper. We have modified the paper according to the reviewers’ comments and provided line by line response to their comments and reference to changes in the revised paper. If the reviewer has new comments on the revised manuscript, we will revise it again.

Replies to the comments of Reviewer # 1

There is a minor issue in the interaction hypothesis development section. Like "H3b: Job resources can reduce the negative effect of job demands on postdoctoral burnout" is erroneously stated. It should be "Job resources can reduce the positive effect of job demands on postdoctoral burnout" as job demands are generally positively associated with the burnout outcomes.

Moreover, I would suggest to cite few lines from the discussion section of the paper referred below to further clarify the coping and gain spiral mechanism for your interaction hypothesis.

Mehboob, F., Othman, N., Fareed, M., & Raza, A. (2022). Change Appraisals and Job Crafting as Foundation to Inculcate Support for Change: A Dual Manifestation. Revista Brasileira de Gestão de Negócios, 24, 207-229.

Reply: We apologize for the mistake and and thank you for the correction. We have corrected H3b to "Job resources can reduce the positive effect of job demands on postdoctoral burnout". For details, please refer to L#34 of page 7. 

Thank you for providing us with valuable references. We have supplemented the relevant content in the discussion section based on the references to clarify the interaction hypothesis. For details, please refer to L#21-24,33-37 of page 18. 

Replies to the comments of Reviewer # 2

Please use the following article to justify the benchmark being met in confirmatory factor analysis for Cronbach's alpha, composite reliability, and AVE values:

Abboh, U. A., Majid, A. H., Fareed, M., & Abdussalaam, I. I. (2022). High-performance work practices lecturers’ performance connection: Does working condition matter? Management in Education, 0(0). https://doi.org/10.1177/08920206211051468

Reply: Thank you for providing us with valuable references. We have supplemented the relevant content in the result section based on the references to justify the benchmark being met . For details, please refer to L#10-11 of page 12, L#10-1 of page 13.

---

## [Editor Report · Decision Letter 2]

18 Oct 2023

Job Demands，Job Resources and Postdoctoral Job Satisfaction：An Empirical Study Based on the Data from 2020 Nature’s Global Postdoc Survey

PONE-D-23-15985R2

Dear Dr. Xinxing Duan,

We’re pleased to inform you that your manuscript has been judged scientifically suitable for publication and will be formally accepted for publication once it meets all outstanding technical requirements.

Kind regards,

Muhammad Fareed, Ph.D

Academic Editor

PLOS ONE

Additional Editor Comments (optional):

Dear Authors,

Thank you for making the corrections as per reviewers' comments.

We are delighted to inform you that your article is accepted.

Thank you.
---

## [Editor Report · Acceptance letter]

20 Oct 2023

PONE-D-23-15985R2 

Job demands，job resources and postdoctoral job satisfaction：an empirical study based on the data from 2020 *Nature* global postdoctoral survey 

Dear Dr. Duan:

I'm pleased to inform you that your manuscript has been deemed suitable for publication in PLOS ONE. Congratulations! Your manuscript is now with our production department. 

Kind regards, 

on behalf of

Dr. Muhammad Fareed 

Academic Editor

PLOS ONE